# Daily evaluation of 26 precipitation datasets using Stage-IV gauge-radar data for the CONUS

Hylke E. Beck[1], Ming Pan[1], Tirthankar Roy[1], Graham P. Weedon[2], Florian Pappenberger[3], Albert I. J. M. van Dijk[4], George J. Huffman[5], Robert F. Adler[6], and Eric F. Wood[1]

[1]Department of Civil and Environmental Engineering, Princeton University, Princeton, New Jersey, USA
[2]Met Office, JCHMR, Maclean Building, Benson Lane, Crowmarsh Gifford, Oxfordshire, UK
[3]European Centre for Medium-Range Weather Forecasts (ECMWF), Reading, UK
[4]Fenner School for Environment and Society, Australian National University, Canberra, Australia
[5]NASA Goddard Space Flight Center (GSFC), Greenbelt, Maryland, USA
[6]University of Maryland, Earth System Science Interdisciplinary Center, College Park, Maryland, USA

**Correspondence:** Hylke E. Beck (hylke.beck@gmail.com)

**Abstract.**

New precipitation ($P$) datasets are released regularly, following innovations in weather forecasting models, satellite retrieval methods, and multi-source merging techniques. Using the conterminous US as a case study, we evaluated the performance of 26 gridded (sub-)daily $P$ datasets to obtain insight in the merit of these innovations. The evaluation was performed at a daily timescale for the period 2008–2017 using the Kling-Gupta Efficiency (KGE), a performance metric combining correlation, bias, and variability. As reference, we used the high-resolution (4 km) Stage-IV gauge-radar $P$ dataset. Among the three KGE components, the $P$ datasets performed worst overall in terms of correlation (related to event identification). In terms of improving KGE scores for these datasets, improved $P$ totals (affecting the bias score) and improved distribution of $P$ intensity (affecting the variability score) are of secondary importance. Among the 11 gauge-corrected $P$ datasets, the best overall performance was obtained by MSWEP V2.2, underscoring the importance of applying daily gauge corrections and accounting for gauge reporting times. Several uncorrected $P$ datasets outperformed gauge-corrected ones. Among the 15 uncorrected $P$ datasets, the best performance was obtained by the fourth-generation reanalysis ERA5-HRES, reflecting the significant advances in earth system modeling during the last decade. The (re)analyses generally performed better in winter than in summer, while the opposite was the case for the satellite-based datasets. IMERGHH V05 performed substantially better than TMPA-3B42RT V7, attributable to the many improvements implemented in the IMERG satellite $P$ retrieval algorithm. IMERGHH V05 outperformed ERA5-HRES in regions dominated by convective storms, while the opposite was observed in regions of complex terrain. The ERA5-EDA ensemble average exhibited higher correlations than the ERA5-HRES deterministic run, highlighting the value of ensemble modeling. The regional convection-permitting climate model WRF showed considerably more accurate $P$ totals over the mountainous west and performed best among the uncorrected datasets in terms of variability, suggesting there is merit in using high-resolution models to obtain climatological $P$ statistics. Our findings ~~can be used as a guide~~ provide some guidance to choose the most suitable $P$ dataset for a particular application.

# 1 Introduction

Knowledge about the spatio-temporal distribution of precipitation ($P$) is important for a multitude of scientific and operational applications, including flood forecasting, agricultural monitoring, and disease tracking (Tapiador et al., 2012; Kucera et al., 2013; Kirschbaum et al., 2017). However, $P$ is highly variable in space and time and therefore extremely challenging to esti-
5 mate, especially in topographically complex, convection-dominated, and snowfall-dominated regions ~~(Stephens et al., 2010; Herold et al., 2~~ (Stephens et al., 2010; Tian and Peters-Lidard, 2010; Herold et al., 2016; Prein and Gobiet, 2017). In the past decades, numerous gridded $P$ datasets have been developed, differing in terms of design objective, spatio-temporal resolution and coverage, data sources, algorithm, and latency (see Tables 1 and 2 for an overview of quasi- and fully-global datasets).

A large number of regional-scale studies have evaluated gridded $P$ datasets to obtain insight in the merit of different methods
and innovations (see reviews by Gebremichael, 2010, Maggioni et al., 2016, and Sun et al., 2018). However, many of these studies: (i) used only a subset of the available $P$ datasets, and omitted (re)analyses, which have higher skill in cold periods and regions (Huffman et al., 1995; Ebert et al., 2007; Beck et al., 2017c); (ii) focused on a small (sub-continental) region, limiting the generalizability of the findings; (iii) considered a small number ($< 50$) of rain gauges or streamflow gauging stations for the evaluation, limiting the validity of the findings; (iv) used gauge observations already incorporated in the datasets as reference
without explicitly mentioning this, potentially leading to a biased evaluation; and (v) failed to account for gauge reporting times, possibly resulting in spurious temporal mismatches between the datasets and the gauge observations.

In an effort to obtain more generally valid conclusions, we recently evaluated 22 (sub-)daily gridded $P$ datasets using gauge observations ($\sim 75\,000$ stations) and hydrological modeling ($\sim 9000$ catchments) globally (Beck et al., 2017c). Other noteworthy large-scale assessments include Tian and Peters-Lidard (2010), who quantified the uncertainty in $P$ estimates by
20 comparing six satellite-based datasets, Massari et al. (2017), who evaluated five $P$ datasets using triple collocation at the daily time scale without the use of ground observations, and Sun et al. (2018), who compared 19 $P$ datasets ~~from~~ at daily to annual timescales. These comprehensive studies highlighted (among other things): (i) substantial differences among $P$ datasets and thus the importance of dataset choice; (ii) the complementary strengths of satellite and (re)analysis $P$ datasets; (iii) the value of merging $P$ estimates from disparate sources; (iv) the effectiveness of daily (as opposed to monthly) gauge corrections; and
25 (v) the widespread underestimation of $P$ in mountainous regions.

Here, we evaluate an even larger selection of (sub-)daily (quasi-)global $P$ datasets for the conterminous US (CONUS), including some promising recently released datasets: ERA5 (the successor to ERA-Interim; ~~Hersbach and Dee, 2016~~Hersbach et al., 2018 ), IMERG (the successor to TMPA; Huffman et al., 2014, 2018), and MERRA-2 (one of the few reanalysis $P$ datasets incorporating daily gauge observations; Gelaro et al., 2017; Reichle et al., 2017). In addition, we evaluate the performance of a
30 regional convection-permitting climate model (WRF; Liu et al., 2017). As reference, we use the high-resolution, radar-based, gauge-adjusted Stage-IV $P$ dataset (Lin and Mitchell, 2005) produced by the National Centers for Environmental Prediction (NCEP). As performance metric, we adopt the widely used Kling-Gupta Efficiency (KGE; Gupta et al., 2009; Kling et al., 2012). We shed light on the strengths and weaknesses of different $P$ datasets and on the merit of different technological and methodological innovations by addressing ~~nine~~ ten pertinent questions:

1. What is the most important factor determining a high KGE score?

2. How do the uncorrected $P$ datasets perform?

3. How do the gauge-based $P$ datasets perform?

4. How do the $P$ datasets perform in summer versus winter?

5. What is the impact of gauge corrections?

6. What is the improvement of IMERG over TMPA?

7. What is the improvement of ERA5 over ERA-Interim?

8. How does the ERA5-EDA ensemble average compare to the ERA5-HRES deterministic run?

9. How do IMERG and ERA5 compare?

10. How well does a regional convection-permitting climate model perform?

## 2   Data and methods

### 2.1   $P$ datasets

We evaluated the performance of 26 gridded (sub-)daily $P$ datasets (Tables 1 and 2). All datasets are either fully- or near-global with the exception of WRF, which is limited to the CONUS. The datasets are classified as either uncorrected, which implies that temporal variations depend entirely on satellite and/or (re)analysis data, or corrected, which implies that temporal variations depend to some degree on gauge observations. We included seven datasets exclusively based on satellite data (CMORPH V1.0, GSMaP-Std V6, IMERGHHE V05, PERSIANN, PERSIANN-CCS, SM2RAIN-CCI V2, and TMPA-3B42RT V7), six fully based on (re)analyses (ERA-Interim, ERA5-HRES, ERA5-EDA, GDAS-Anl, JRA-55, and NCEP-CFSR; although ERA5 assimilates radar and gauge data over the CONUS), one incorporating both satellite and (re)analysis data (CHIRP V2.0), and one based on a regional convection-permitting climate model (WRF).

Among the gauge-based $P$ datasets, six combined gauge and satellite data (CMORPH-CRT V1.0, GPCP-1DD V1.2, GSMaP-Std Gauge V7, IMERGDF V05, PERSIANN-CDR V1R1, and TMPA-3B42 V7), one combined gauge and reanalysis data (WFDEI-GPCC), three combined gauge, satellite, and (re)analysis data (CHIRPS V2.0, MERRA-2 and MSWEP V2.2), while one was fully based on gauge observations (CPC Unified V1.0/RT). For transparency and reproducibility, we report dataset version numbers throughout the study for the datasets for which this information was provided. For the $P$ datasets with a sub-daily temporal resolution, we calculated daily accumulations for 00:00–23:59 UTC. $P$ datasets with spatial resolutions $< 0.1°$ were resampled to $0.1°$ using bilinear averaging, whereas those with spatial resolutions $> 0.1°$ were resampled to $0.1°$ using bilinear interpolation.

## 2.2 Stage-IV gauge-radar data

As reference, we used the NCEP Stage-IV dataset, which has a 4-km spatial and hourly temporal resolution and covers the period 2002 until the present, and merges data from 140 radars and ∼5500 gauges ~~(Lin and Mitchell, 2005)~~ over the CONUS (Lin and Mitchell, 2005). Stage-IV provides highly accurate $P$ estimates and has therefore been widely used as reference for
the evaluation of $P$ datasets (e.g., Hong et al., 2006; Habib et al., 2009; AghaKouchak et al., 2011, 2012; Nelson et al., 2016; Zhang et al., 2018). Daily Stage-IV data are available but they represent an accumulation period that is incompatible with the datasets we are evaluating (12:00–11:59 UTC instead of 00:00–23:59 UTC). We therefore calculated daily accumulations for 00:00–23:59 UTC from 6-hourly Stage-IV accumulations. The Stage-IV dataset was reprojected from its native 4-km polar stereographic projection to a regular geographic $0.1°$ grid using bilinear averaging.

The Stage-IV dataset is a mosaic of regional analyses produced by 12 CONUS River Forecast Centers (RFCs) and is thus subject to the gauge correction and quality control performed at each individual RFC ~~(Westrick et al., 1999; Eldardiry et al., 2017)~~ (Westrick et al., 1999; Smalley et al., 2014; Eldardiry et al., 2017). To reduce systematic biases, the Stage-IV dataset was rescaled such that its long-term mean matches that of the PRISM dataset (Daly et al., 2008) for the evaluation period (2008–2017). To this end, the PRISM dataset was upscaled from ∼800 m to $0.1°$ using bilinear averaging. The PRISM dataset has been derived
from gauge observations using a sophisticated interpolation approach that accounts for topography. It is generally considered the most accurate monthly $P$ dataset available for the US and has been used as reference in numerous studies (e.g., Mizukami and Smith, 2012; Prat and Nelson, 2015; Liu et al., 2017). However, the dataset has not been corrected for wind-induced gauge undercatch and thus may underestimate $P$ to some degree (Groisman and Legates, 1994; Rasmussen et al., 2012).

## 2.3 Evaluation approach

The evaluation was performed at a daily ~~temporal~~ temporal and $0.1°$ ~~spatial-resolution~~ spatial resolution by calculating, for each grid-cell, Kling-Gupta Efficiency (KGE) scores from daily time series for the ten-year period from 2008 to 2017. KGE is an objective performance metric combining correlation, bias, and variability. It was introduced in Gupta et al. (2009) and modified in Kling et al. (2012) and is defined as follows:

$$\text{KGE} = 1 - \sqrt{(r-1)^2 + (\beta-1)^2 + (\gamma-1)^2}, \tag{1}$$

where the correlation component $r$ is represented by ~~the~~ (Pearson's) ~~coefficient of correlation~~ correlation coefficient, the bias component $\beta$ by the ratio of estimated and observed means, and the variability component $\gamma$ by the ratio of the estimated and observed coefficients of variation:

$$\beta = \frac{\mu_s}{\mu_o} \quad \text{and} \quad \gamma = \frac{\sigma_s/\mu_s}{\sigma_o/\mu_o}, \tag{2}$$

where $\mu$ and $\sigma$ are the distribution mean and standard deviation, respectively, and the subscripts $s$ and $o$ indicate estimate and
reference, respectively. KGE, $r$, $\beta$, and $\gamma$ values all have their optimum at unity. The correlation is primarily sensitive to the dynamics of $P$ extremes, while the bias and variability ratio are mainly sensitive to the magnitude of more frequent events.

**Table 1.** Overview of the 15 uncorrected (quasi-)global (sub-)daily gridded $P$ datasets evaluated in this study. The 11 gauge-corrected datasets are listed in Table 2. Abbreviations in the data source(s) column defined as: S, satellite; R, reanalysis; A, analysis; and M, regional climate model. The acronym NRT in the temporal coverage column stands for Near Real-Time. In the spatial coverage column, "Global" means fully global coverage including oceans, while "Land" means that the coverage is limited to the terrestrial land surface.

| Name | Details | Data source(s) | Spatial resolution | Spatial coverage | Temporal resolution | Temporal coverage | Reference or website |
|---|---|---|---|---|---|---|---|
| CHIRP V2.0[1] | Climate Hazards group InfraRed Precipitation (CHIRP) V2.0 | S, R, A | 0.05° | Land, 50°N/S | Daily | 1981–NRT[3] | Funk et al. (2015a) |
| CMORPH V1.0 | CPC MORPHing technique (CMORPH) V1.0 | S | 0.07° | 60°N/S | 30 minutes | 1998–NRT[2] | Joyce et al. (2004); Xie et al. (2017) |
| ERA-Interim | European Centre for Medium-range Weather Forecasts ReAnalysis Interim (ERA-Interim) | R | ∼0.75° | Global | 3-hourly | 1979–NRT[5] | Dee et al. (2011) |
| ERA5-HRES[5] | European Centre for Medium-range Weather Forecasts ReAnalysis 5 (ERA5) High RESolution (HRES) | R | ∼0.28° | Global | Hourly | 2008–NRT[3,5] | ~~Hersbach and Dee (2016)~~ Hersbach et al. (2018) |
| ERA5-EDA[5] | European Centre for Medium-range Weather Forecasts ReAnalysis 5 (ERA5) Ensemble Data Assimilation (EDA) ensemble mean | R | ∼0.56° | Global | Hourly | 2008–NRT[3,5] | ~~Hersbach and Dee (2016)~~ Hersbach et al. (2018) |
| GDAS-Anl | National Centers for Environmental Prediction (NCEP) Global Data Assimilation System (GDAS) Analysis (Anl) | A | ∼0.25° | Global | 3-hourly | 2015–NRT[2] | www.emc.ncep.noaa.gov/gmb/gdas/ |
| GSMaP-Std V6 | Global Satellite Mapping of Precipitation (GSMaP) Moving Vector with Kalman (MVK) Standard V6 | S | 0.1° | 60°N/S | Hourly | 2000–NRT[2] | Ushio et al. (2009) |
| IMERGHHE V05 | Integrated Multi-satellitE Retrievals for GPM (IMERG) early run V05 | S | 0.1° | 60°N/S | 30 minutes | 2014–NRT[1,6] | Huffman et al. (2014, 2018) |
| JRA-55 | Japanese 55-year ReAnalysis (JRA-55) | R | ∼0.56° | Global | 3-hourly | 1959–NRT[3] | Kobayashi et al. (2015) |
| NCEP-CFSR | National Centers for Environmental Prediction (NCEP) Climate Forecast System Reanalysis (CFSR) | R | ∼0.31° | Global | Hourly | 1979–2010 | Saha et al. (2010) |
| PERSIANN | Precipitation Estimation from Remotely Sensed Information using Artificial Neural Networks (PERSIANN) | S | 0.25° | 60°N/S | Hourly | 2000–NRT[2] | Sorooshian et al. (2000) |
| PERSIANN-CCS | Precipitation Estimation from Remotely Sensed Information using Artificial Neural Networks (PERSIANN) Cloud Classification System (CCS) | S | 0.04° | 60°N/S | Hourly | 2003–NRT[2] | Hong et al. (2004) |
| SM2RAIN-CCI V2 | Rainfall inferred from European Space Agency's (ESA) Climate Change Initiative (CCI) satellite near-surface soil moisture V2 | S | 0.25° | Land | Daily | 1998–2015 | Ciabatta et al. (2018) |
| TMPA-3B42RT V7 | TRMM Multi-satellite Precipitation Analysis (TMPA) 3B42RT V7 | S | 0.25° | 50°N/S | 3-hourly | 2000–NRT[2] | Huffman et al. (2007) |
| WRF[8] | Weather Research and Forecasting (WRF) | M | 4 km | CONUS | Hourly | 2000–2013 | Liu et al. (2017) |

[1] The day-to-day variability was based entirely on satellite and reanalysis data. However, the monthly climatology was corrected using a gauge-based dataset (Funk et al., 2015b).

[2] Available until the present with a delay of several hours.

[3] Available until the present with a delay of several days.

[4] Available until the present with a delay of several months.

[5] Rain gauge and ground radar observations were assimilated from 17 July 2009 onwards (Lopez, 2011, 2013).

[6] 1950–NRT once production has completed.

[7] 2000–NRT for the next version.

[8] The only dataset included in the evaluation with continental coverage instead of (quasi-)global coverage.

**Table 2.** Overview of the 11 gauge-corrected (quasi-)global (sub-)daily gridded $P$ datasets evaluated in this study. The 15 uncorrected datasets are listed in Table 1. Abbreviations in the data source(s) column defined as: G, gauge; S, satellite; and R, reanalysis; and A, analysis. The acronym NRT in the temporal coverage column stands for Near Real-Time. In the spatial coverage column, "global" indicates fully global coverage including ocean areas, while "land" indicates that the coverage is limited to the terrestrial surface.

| Name | Details | Data source(s) | Spatial resolution | Spatial coverage | Temporal resolution | Temporal coverage | Reference or website |
|---|---|---|---|---|---|---|---|
| CHIRPS V2.0 | Climate Hazards group InfraRed Precipitation with Stations (CHIRPS) V2.0 | G, S, R, A | 0.05° | Land, 50°N/S | Daily | 1981–NRT[2] | Funk et al. (2015a) |
| CMORPH-CRT V1.0 | CPC MORPHing technique (CMORPH) bias corrected (CRT) V1.0 | G, S | 0.07° | 60°N/S | 30 minutes | 1998–2015 | Joyce et al. (2004); Xie et al. (2017) |
| CPC Unified V1.0/RT | Climate Prediction Center (CPC) Unified V1.0 and RT | G | 0.5° | Land | Daily | 1979–NRT[2] | Xie et al. (2007); Chen et al. (2008) |
| GPCP-1DD V1.2 | Global Precipitation Climatology Project (GPCP) 1-Degree Daily (1DD) Combination V1.2 | G, S | 1° | Global | Daily | 1996–2015 | Huffman et al. (2001) |
| GSMaP-Std Gauge V7 | Global Satellite Mapping of Precipitation (GSMaP) Moving Vector with Kalman (MVK) Standard gauge-corrected V7 | G, S | 0.1° | 60°N/S | Hourly | 2000–NRT[1] | Ushio et al. (2009) |
| IMERGDF V05 | Integrated Multi-satellitE Retrievals for GPM (IMERG) final run V05 | G, S | 0.1° | 60°N/S | 30 minutes | 2014–NRT[3, 4] | Huffman et al. (2014, 2018) |
| MERRA-2 | Modern-Era Retrospective Analysis for Research and Applications 2 | G, S, R | ~0.5° | Global | Hourly | 1980–NRT[3] | Gelaro et al. (2017); Reichle et al. (2017) |
| MSWEP V2.2 | Multi-Source Weighted-Ensemble Precipitation (MSWEP) V2.2 | G, S, R, A | 0.1° | Global | 3-hourly | 1979–NRT[1] | Beck et al. (2017b, 2019) |
| PERSIANN-CDR V1R1 | Precipitation Estimation from Remotely Sensed Information using Artificial Neural Networks (PERSIANN) Climate Data Record (CDR) V1R1 | G, S | 0.25° | 60°N/S | Daily | 1983–2016 | Ashouri et al. (2015) |
| TMPA-3B42 V7 | TRMM Multi-satellite Precipitation Analysis (TMPA) 3B42 V7 | G, S | 0.25° | 50°N/S | 3-hourly | 2000–2017 | Huffman et al. (2007) |
| WFDEI-GPCC | WATCH Forcing Data ERA-Interim (WFDEI) corrected using Global Precipitation Climatology Centre (GPCC) | G, R | 0.5° | Land | 3-hourly | 1979–2016 | Weedon et al. (2014) |

[1] Available until the present with a delay of several hours.

[2] Available until the present with a delay of several days.

[3] Available until the present with a delay of several months.

[4] 2000–NRT for the next version.

## 3 Results and Discussion

### 3.1 What is the most important factor determining a high KGE score?

Figure 2 presents box-and-whisker plots of KGE scores for the 26 $P$ datasets. The mean median KGE score over all datasets is 0.54. The mean median scores for the correlation, bias, and variability components of the KGE, expressed as $|r - 1|$, $|\beta - 1|$, and $|\gamma - 1|$, are $-0.34$, $-0.18$, and $-0.16$, respectively (see Equation 1). The datasets thus performed considerably worse in terms of correlation, which makes sense given that long-term climatological $P$ statistics are easier to estimate than day-to-day $P$ dynamics. Due to the squaring of the three components in the KGE equation (see Equation 1), the correlation values exert the dominant influence on on the final KGE scores. Indeed, the performance ranking in terms of KGE corresponds well with the performance ranking in terms of correlation (Figure 2). These results suggest that in order to get an improved KGE score the most important component score to improve is the correlation. This in turn suggests that, for existing daily $P$ datasets, improvements to the timing of $P$ events at the daily scale (dominating the correlation scores) are more valuable than improvements to $P$ totals (dominating bias scores) or the intensity distribution (dominating variability scores).

### 3.2 How do the uncorrected $P$ datasets perform?

Among the uncorrected $P$ datasets, the (re)analyses performed better overall than the satellite-based datasets (Figures 1 and 2). The best performance was obtained by ECMWF's fourth-generation reanalysis ERA5-HRES (median KGE of 0.63), with NASA's most recent satellite-based dataset IMERGHHE V05 and the ensemble average ERA5-EDA coming a close equal second (median KGE of 0.62). These results underscore the substantial advances in earth system modeling and satellite-based $P$ estimation over the last decade. The third-generation, coarser-resolution reanalyses (ERA-Interim, JRA-55, and NCEP-CFSR) performed slightly worse overall (median KGE of 0.55, 0.52, and 0.52, respectively). ERA-Interim performed slightly better than other third-generation reanalyses, consistent with earlier studies focusing on $P$ (Bromwich et al., 2011; Peña Arancibia et al., 2013; Palerme et al., 2017; Beck et al., 2017c) and other atmospheric variables (Bracegirdle and Marshall, 2012; Jin-Huan et al., 2014; Zhang et al., 2016). All (re)analyses, including the new ERA5-HRES, underestimated the variability (Figure 2 and Supplement Figure S3), reflecting the tendency of (re)analyses to overestimate $P$ frequency (Zolina et al., 2004; Sun et al., 2006; Lopez, 2007; Stephens et al., 2010; Skok et al., 2015; Beck et al., 2017c). The additional variability underestimation by ERA5-EDA compared to ERA5-HRES probably reflects the variance loss induced by the averaging.

Among the uncorrected satellite-based $P$ datasets, the new IMERGHHE V05 performed best overall by a substantial margin (median KGE of 0.62; Figure 1 and 2), reflecting the quality of the new IMERG $P$ retrieval algorithm (Huffman et al., 2014, 2018). The other passive microwave-based datasets (CMORPH V1.0, GSMaP-Std V6, and TMPA-3B42RT V7) obtained median KGE scores ranging from 0.44 to 0.52. CHIRP V2.0, which combines infrared- and reanalysis-based estimates, performed similarly to some of the passive-microwave datasets (median KGE of 0.47). The datasets exclusively based on infrared data (PERSIANN and PERSIANN-CCS) performed markedly worse (median KGE of 0.34 and 0.32, respectively), consistent with previous $P$ dataset evaluations (e.g., Hirpa et al., 2010; Peña Arancibia et al., 2013; Cattani et al., 2016; Beck et al., 2017c). This has been attributed to the indirect nature of the relationship between cloud-top temperatures and surface rainfall (Adler

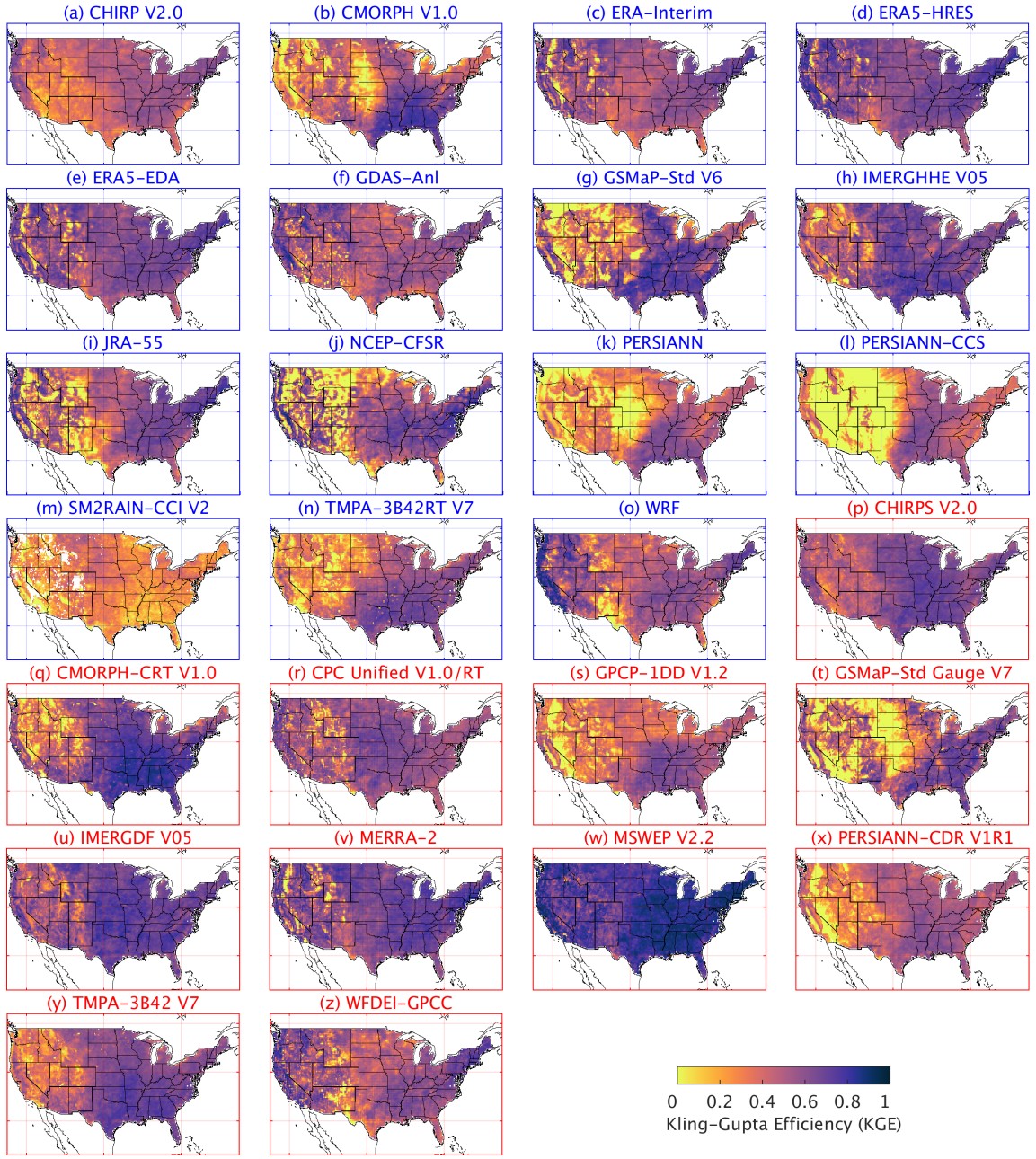

**Figure 1.** Kling-Gupta Efficiency (KGE) scores for the 26 gridded $P$ datasets using the Stage-IV gauge-radar dataset as reference. White indicates missing data. Higher KGE values correspond to better performance. Uncorrected datasets are listed in blue, whereas gauge-corrected datasets are listed in red. Details on the datasets are provided in Tables 1 and 2. Maps for the correlation, bias, and variability components of the KGE are presented in the Supplement.

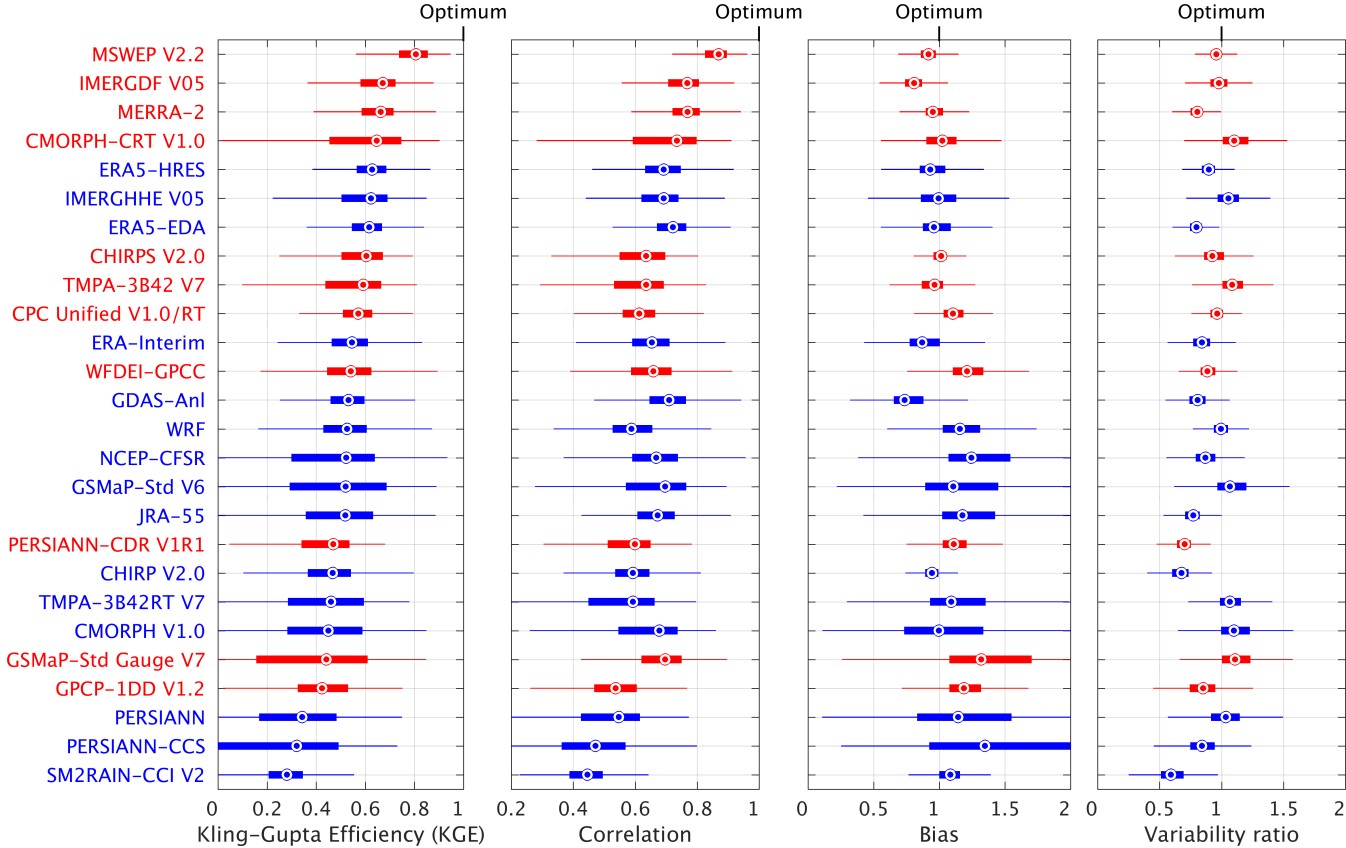

**Figure 2.** Box-and-whisker plots of Kling-Gupta Efficiency (KGE) scores for the 26 gridded $P$ datasets using the Stage-IV gauge-radar dataset as reference. The circles represents the median value, the left and right edges of the box represent the 25th and 75th percentile values, respectively, while the 'whiskers' represent the extreme values. The statistics were calculated for each dataset from the distribution of grid-cell KGE values (no area-weighting was performed). The datasets are sorted in ascending order of the median KGE. Uncorrected datasets are indicated in blue, whereas gauge-corrected datasets are indicated in red. Details on the datasets are provided in Tables 1 and 2.

and Negri, 1988; Vicente et al., 1998; Scofield and Kuligowski, 2003). The infrared-based datasets generally exhibited a much larger spatial variability in performance for all four metrics (Figure 1 and Supplement Figures S1–S3).

The (uncorrected) satellite soil moisture-based SM2RAIN-CCI V2 dataset performed comparatively poorly (median KGE of 0.28; Figures 1 and 2). The dataset strongly underestimated the variability (Supplement Figure S3), due to the noisiness of satellite soil moisture retrievals and the inability of satellite soil moisture-based algorithms to detect rainfall exceeding the soil water storage capacity (Zhan et al., 2015; Wanders et al., 2015; Tarpanelli et al., 2017; Ciabatta et al., 2018). At high latitudes and elevations, the presence of snow and frozen soils may have hampered performance (Brocca et al., 2014), while in arid regions, irrigation may have been misinterpreted as rainfall (Brocca et al., 2018). In addition, approximately 25 % (in the eastern CONUS) to 50 % (over the mountainous west) of the daily rainfall values were based on temporal interpolation, to fill

gaps in the satellite soil moisture data (Dorigo et al., 2017). Despite these limitations, the SM2RAIN datasets may provide new possibilities for evaluation (Massari et al., 2017) and correction (Massari et al., 2018) of other $P$ datasets, since they constitute a fully independent, alternative source of rainfall data.

All uncorrected $P$ datasets exhibited lower overall performance in the western CONUS (Figures 1 and 2, and Supplement Figures S1–S3), in line with previous studies (e.g., ~~Gottschalck et al., 2005; Ebert et al., 2007; Tian et al., 2007; AghaKouchak et al., 2012;~~ Gottschalck et al., 2005; Ebert et al., 2007; Tian et al., 2007; AghaKouchak et al., 2012; Chen et al., 2013; Beck et al., 2017c; Gebregiorgis ). This is attributable to the more complex topography and greater spatio-temporal heterogeneity of $P$ in the west (Daly et al., 2008), which affects the quality of both the evaluated datasets and the reference ~~(Westrick et al., 1999; Eldardiry et al., 2017)~~ (Westrick et al., 1999; Smalley et al., 2014; Eldardiry et al., 2017). With the exception of CHIRP V2.0 (which has been corrected for systematic biases using gauge observations; Funk et al., 2015b) and WRF (the high-resolution climate simulation; Liu et al., 2017), the (uncorrected) datasets exhibited large $P$ biases over the mountainous west (Supplement Figure S2), which is in agreement with earlier studies using other reference datasets (Adam et al., 2006; Kauffeldt et al., 2013; Beck et al., 2017a, c) and reflects the difficulty of retrieving and simulating orographic $P$ (Roe, 2005). We initially expected bias values to be higher than unity since PRISM, the dataset used to correct systematic biases in Stage-IV (see Section 2.2), lacks explicit gauge undercatch corrections (Daly et al., 2008), but this did not appear to be the case (Figure 2 and Supplement Figure S2).

### 3.3 How do the gauge-based $P$ datasets perform?

Among the gauge-based $P$ datasets, the best overall performance was obtained by MSWEP V2.2 (median KGE of 0.81), followed at some distance by IMERGDF V05 (median KGE of 0.67) and MERRA-2 (median KGE of 0.66; Figures 1 and 2). IMERGDF V05 exhibited a small negative bias, while MERRA-2 slightly underestimated the variability. The good performance obtained by MSWEP V2.2 underscores the importance incorporating daily gauge data and accounting for reporting times (Beck et al., 2019). While CMORPH-CRT V1.0, CPC Unified V1.0/RT, GSMaP-Std Gauge V7, and MERRA-2 also incorporate daily gauge data, they did not account for reporting times, resulting in temporal mismatches and hence lower KGE scores (Figure 2). Reporting times in the CONUS range from midnight $-12$ to $+9$ hours UTC for the stations in the comprehensive GHCN-D gauge database (Menne et al., 2012; Figure 2c in Beck et al., 2019), suggesting that ~~for some stations,~~ up to half of the daily $P$ accumulations may be assigned to the wrong day. In addition, CMORPH-CRT V1.0, GSMaP-Std Gauge V7, and MERRA-2 applied daily gauge corrections using CPC Unified (Xie et al., 2007; Chen et al., 2008), which has a relatively coarse 0.5° resolution, whereas MSWEP V2.2 applied corrections at 0.1° resolution based on the five nearest gauges for each grid-cell (Beck et al., 2019). The good performance of IMERGDF V05 is somewhat surprising, given the use of monthly rather than daily gauge data, and attests to the quality of the IMERG $P$ retrieval algorithm (Huffman et al., 2014, 2018).

Similar to the uncorrected datasets, the corrected estimates consistently performed worse in the west (Figures 1 and 2 and Supplement Figures S1–S3), due not only to the greater spatio-temporal heterogeneity in $P$ (Daly et al., 2008) but also the lower gauge network density (Kidd et al., 2017). It should be kept in mind that the performance ranking may differ across the globe depending on the amount of gauge data ingested and the quality control applied for each dataset. Thus, the results found here for the CONUS do not necessarily directly generalize to other regions.

### 3.4 How do the $P$ datasets perform in summer versus winter?

Figure 3 presents KGE values for summer and winter for the 26 $P$ datasets. The following observations can be made:

- The spread in median KGE values among the datasets is much greater in winter than in summer. In addition, almost all datasets exhibit a greater spatial variability in KGE values in winter, as indicated by the wider boxes and whiskers. This is probably at least partly attributable to the lower quality of the Stage-IV dataset in winter (Westrick et al., 1999; Smalley et al., 2014; E .

- All (re)analyses (with the exception of NCEP-CFSR) including the regional climate model WRF consistently performed better in winter than in summer. This is because predictable large-scale stratiform systems dominate in winter (Adler et al., 2001; Eber , whereas unpredictable small-scale convective cells dominate in summer (Arakawa, 2004; Prein et al., 2015).

- All satellite $P$ datasets (with the exception of PERSIANN) consistently performed better in summer than in winter. Satellites are ideally suited to detect the intense, localized convective storms which dominate in summer (Wardah et al., 2008; AghaK . Conversely, there are major challenges associated with the retrieval of snowfall (Kongoli et al., 2003; Liu and Seo, 2013; Skofronick and light rainfall (Habib et al., 2009; Kubota et al., 2009; Tian et al., 2009; Lu and Yong, 2018), affecting the performance in winter.

- The datasets incorporating both satellite and reanalysis estimates (CHIRP V2.0, CHIRPS V2.0, and MSWEP V2.2) performed similarly in both seasons, taking advantage of the accuracy of satellite retrievals in summer and reanalysis outputs in winter (Ebert et al., 2007; Beck et al., 2017b). The fully gauge-based CPC Unified V1.0/RT also performed similarly in both seasons.

### 3.5 What is the impact of gauge corrections?

Differences in median KGE values between uncorrected and gauge-corrected versions of $P$ datasets ranged from $-0.07$ (GSMaP-Std Gauge V7) to $+0.20$ (CMORPH-CRT V1.0; Table 3). GSMaP-Std Gauge V7 shows a large positive bias in the west (Supplement Figure S2), suggesting that its gauge-correction methodology requires re-evaluation. The substantial improvements in median KGE for CHIRPS V2.0 ($+0.13$) and CMORPH-CRT V1.0 ($+0.20$) reflect the use of sub-monthly gauge data (5-day and daily, respectively). Conversely, the datasets incorporating monthly gauge data (IMERGDF V05 and WFDEI-GPCC) exhibited little to no improvement in median KGE ($+0.05$ and $-0.01$, respectively), suggesting that monthly corrections provide little to no benefit at the daily timescale of the present evaluation ~~adn that the error is dominated by errors in teh diurnal cycle~~(Tan and Santo, 2018). These results, combined with the fact that several uncorrected $P$ datasets outperformed gauge-corrected ones (Figure 2), suggest that a $P$ dataset labeled as "gauge-corrected" is not necessarily always the better choice. ~~The local density of gauge observations and the typical scale of local $P$ systems need to be considered when evaluating this issue.~~

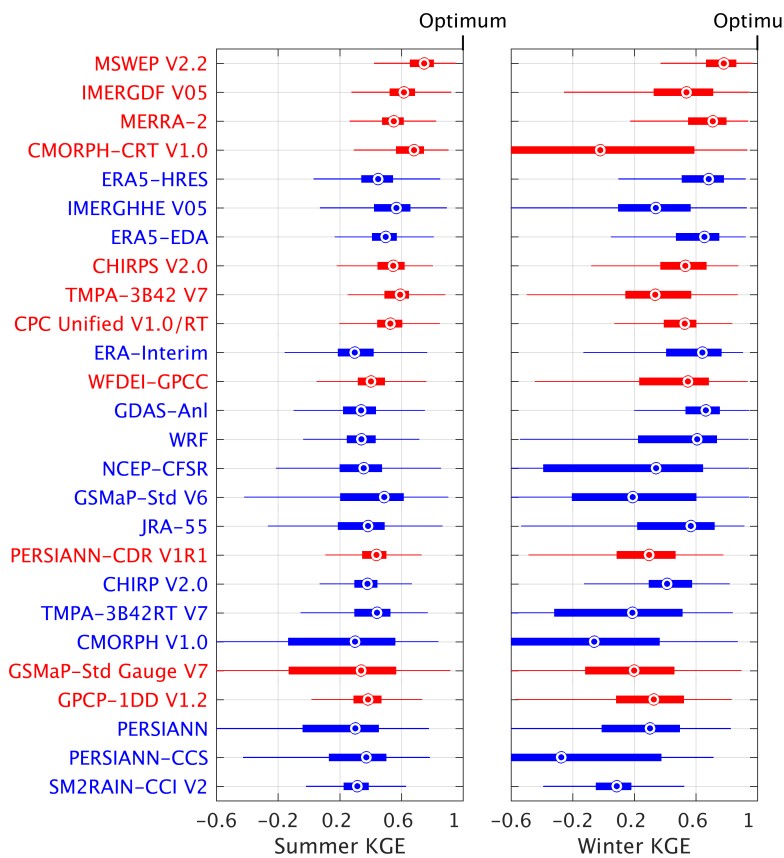

**Figure 3.** Box-and-whisker plots of Kling-Gupta Efficiency (KGE) scores for summer (June–August) and winter (December–February) using the Stage-IV gauge-radar dataset as reference. The circles represents the median value, the left and right edges of the box represent the 25th and 75th percentile values, respectively, while the 'whiskers' represent the extreme values. The statistics were calculated for each dataset from the distribution of grid-cell KGE values (no area-weighting was performed). The datasets are sorted in ascending order of the overall median KGE (see Figure 2). Uncorrected datasets are indicated in blue, whereas gauge-corrected datasets are indicated in red. Details on the datasets are provided in Tables 1 and 2.

**Table 3.** Difference in median Kling-Gupta Efficiency (KGE) between uncorrected and gauge-corrected versions of $P$ datasets. Tables 1 and 2 provide details of the datasets.

| Uncorrected dataset | Corrected dataset | $\Delta\overline{\text{KGE}}$ | Correction approach | Reference |
|---|---|---|---|---|
| IMERGHHE V05 | IMERGDF V05 | +0.05 | Monthly corrections using GPCC | Huffman et al. (2018) |
| CHIRP V2.0 | CHIRPS V2.0 | +0.13 | 5-day corrections using compiled database | Funk et al. (2015a) |
| CMORPH V1.0 | CMORPH-CRT V1.0 | +0.20 | Daily corrections using CPC Unified | Xie et al. (2017) |
| ERA-Interim | WFDEI-GPCC | −0.01 | Monthly corrections using GPCC | Weedon et al. (2014) |
| GSMaP-Std V6 | GSMaP-Std Gauge V7 | −0.07 | Daily corrections using CPC Unified | Mega et al. (2014) |

## 3.6    What is the improvement of IMERG over TMPA?

IMERG (Huffman et al., 2014, 2018) is NASA's latest satellite $P$ dataset and is foreseen to replace the TMPA dataset (Huffman et al., 2007; Table 1). The following main improvements were implemented in IMERG compared to TMPA: (i) forward and backward propagation of passive microwave data using CMORPH-style motion vectors (Joyce et al., 2004); (ii) infrared-based rainfall estimates derived using the PERSIANN-CCS algorithm (Hong et al., 2004); (iii) calibration of passive microwave-based $P$ estimates to the Combined GMI-DPR $P$ dataset (available up to almost 70° latitude) during the GPM era and the Combined TMI-PR $P$ dataset (available up to 40° latitude) during the TRMM era; (iv) adjustment of the Combined estimates by GPCP monthly climatological values (Adler et al., 2018) to ameliorate low biases at high latitudes; (v) merging of infrared- and passive microwave-based $P$ estimates using a CMORPH-style Kalman filter; (vi) use of passive microwave data from recent instruments (DMSP-F19, GMI, and NOAA-20); (vii) a 30-minutes temporal resolution (instead of 3-hourly); (viii) a 0.1° spatial resolution (instead of 0.25°); and (ix) greater coverage (essentially complete up to 60° instead of 50° latitude).

These changes have resulted in considerable performance improvements: IMERGHH V05 performed better overall than TMPA-3B42RT V7 in terms of median KGE (0.62 versus 0.46), correlation (0.69 versus 0.59), bias (0.99 versus 1.09), and variability (1.05 versus 1.07; Figures 1, 2, and 4a). The improvement is particularly pronounced over the northern Great Plains (Figure 4a), where TMPA-3B42RT V7 exhibits a large positive bias (Supplement Figure S2). ~~Thus,~~ In the west, however, there are still some small regions over which TMPA-3B42RT V7 performs better (Figure 4a). Overall, our results indicate that there is considerable merit in using IMERGHHE V05 instead of TMPA-3B42RT V7 over the CONUS. Previous studies comparing (different versions of) the same two datasets ~~for~~ over the CONUS (Gebregiorgis et al., 2018), ~~southeast China (Tang et al., 2016b)~~ Bolivia (Satgé et al., 2017), mainland China (Tang et al., 2016a), southeast China (Tang et al., 2016b), Iran (Sharifi et al., 2016), India (Prakash et al., 2016), the Mekong River Basin (Wang et al., 2017), the Tibetan Plateau (Ran et al., 2017), and the northern Andes (Manz et al., 2017) reached largely similar conclusions.

## 3.7    What is the improvement of ERA5 over ERA-Interim?

ERA5 ~~(Hersbach and Dee, 2016)~~ (Hersbach et al., 2018) is ECMWF's recently released fourth-generation reanalysis and the successor to ERA-Interim, generally considered the most accurate third-generation reanalysis (Bromwich et al., 2011; Bracegirdle and Marshall, 2012; Jin-Huan et al., 2014; Beck et al., 2017c; Table 1). ERA5 features several improvements over ERA-Interim, such as: (i) a more recent model and data assimilation system (IFS Cycle 41r2 from 2016 versus IFS Cycle 31r2 from 2006), including numerous improvements in model physics, numerics, and data assimilation; (ii) a higher horizontal resolution (∼0.28° versus ∼0.75°); (iii) more vertical levels (137 versus 60); (iv) assimilation of substantially more observations, including gauge (Lopez, 2013) and ground radar (Lopez, 2011) $P$ data (from 17 July 2009 onwards); (v) a longer temporal span once production has completed (1950–present versus 1979–present) and a near real-time release of the data; (vi) outputs with a higher temporal resolution (hourly versus 3-hourly); and (vii) corresponding uncertainty estimates.

As a result of these changes ERA5-HRES performed markedly better than ERA-Interim in terms of $P$ across most of the CONUS, especially in the west (Figures 1 and 4b). ERA5-HRES obtained a median KGE of 0.63, whereas ERA-Interim

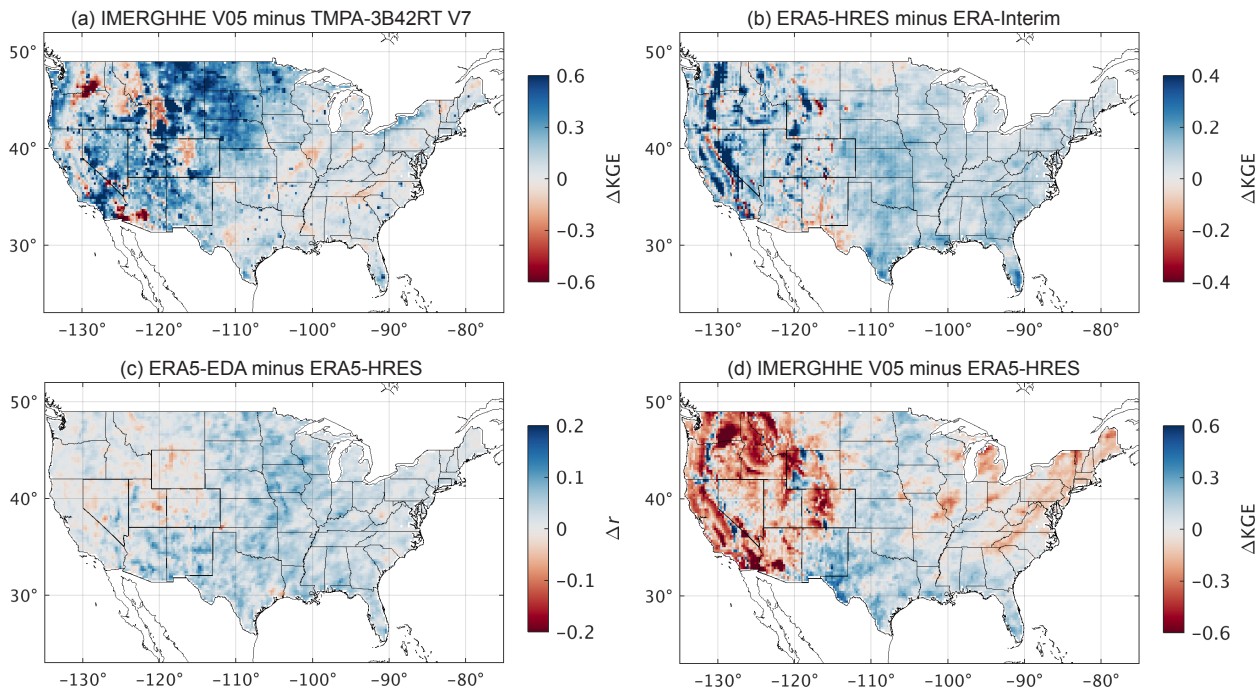

**Figure 4.** (a) Kling-Gupta Efficiency (KGE) scores obtained by IMERGHHE V05 minus those obtained by TMPA-3B42RT V7. (b) KGE scores obtained by ERA5-HRES minus those obtained by ERA-Interim. (c) correlations ($r$) obtained by ERA5-EDA minus those obtained by ERA5-HRES. (d) KGE scores obtained by IMERGHHE V05 minus those obtained by ERA5-HRES. Note the different color scales. The Stage-IV gauge-radar dataset was used as reference. The KGE and correlation values were calculated from daily time series.

obtained a median KGE of 0.55 (Figure 2). Improvements were evident for all three KGE components (correlation, bias, and variability). So far, only two other studies compared the performance of ERA5 and ERA-Interim. The first study compared the two datasets for the CONUS by using them to drive a land surface model (Albergel et al., 2018). The simulations using ERA5 provided substantially better evaporation, soil moisture, river discharge, and snow depth estimates. The authors attributed this

5   to the improved $P$ estimates, which is supported by our results. The second study compared incoming shortwave radiation estimates from ERA5 and ERA-Interim globally, and found that ERA5 provides superior performance (Urraca et al., 2018).

It is difficult to say how much of the performance improvement of ERA5 is due to the assimilation of gauge and radar $P$ data. We suspect that the performance improvement is largely attributable to other factors, given that: (i) the impact of the $P$ data assimilation is limited overall due to the large amount of other observations already assimilated (Lopez, 2013); (ii) radar

10  data were discarded west of 105°W for quality reasons (Lopez, 2011); and (iii) performance improvements were also found in regions without assimilated gauge observations (e.g., Nevada; Figure 4b; Lopez, 2013, their Figure 3). Nevertheless, we expect the performance difference between ERA5 and ERA-Interim to be less in regions with fewer or no assimilated gauge observations (i.e., outside the US, Canada, Argentina, Europe, Iran, and China; Lopez, 2013, their Figure 3).

### 3.8 How does the ERA5-EDA ensemble average compare to the ERA5-HRES deterministic run?

Ensemble modeling involves using outputs from multiple models or from different realizations of the same model; it is widely used in climate, atmospheric, hydrological, and ecological sciences to improve accuracy and quantify uncertainty (Gneiting and Raftery, 2005; Nikulin et al., 2012; Strauch et al., 2012; Cheng et al., 2012; Beck et al., 2013, 2017a). Here, we compare the $P$ estimation performance of a high-resolution ($\sim$0.28°) deterministic reanalysis (ERA5-HRES) to that of a reduced-resolution ($\sim$0.56°) ensemble-average (ERA5-EDA; Table 1). The ensemble consists of ten members generated by perturbing the assimilated observations (Zuo et al., 2017) as well as the model physics (Ollinaho et al., 2016; Leutbecher et al., 2017). The ensemble average was derived by equal weighting of the members.

Compared to ERA5-HRES, we found ERA5-EDA to perform similarly in terms of median KGE (0.62 versus 0.63), better in terms of median correlation (0.72 versus 0.69) and bias (0.96 versus 0.93), but worse in terms of median variability (0.80 versus 0.90; Figures 1, 2, and 4c). The deterioration of the variability is probably at least partly due to averaging, which shifts the distribution toward medium-sized events~~and smoothes any signal~~. The improvement in correlation is evident over the entire CONUS (Figure 4c), and corresponds to a 9 % overall increase in the explained temporal variance, demonstrating the value of ensemble modeling. We expect the improvement to increase with increasing diversity among ensemble members (Brown et al., 2005; DelSole et al., 2014).

### 3.9 How do IMERG and ERA5 compare?

IMERGHHE V05 (Huffman et al., 2014, 2018) and ERA5-HRES ~~(Hersbach and Dee, 2016)~~ (Hersbach et al., 2018) represent the state-of-the-art in terms of satellite $P$ retrieval and reanalysis, respectively (Table 1). Although the datasets exhibited similar performance overall (median KGE of 0.62 and 0.63, respectively; Figures 1 and 2), regionally there were considerable differences (Figure 4d). Compared to ERA5-HRES, IMERGHHE V05 performed substantially worse over regions of complex terrain (including the Rockies and the Appalachians), in line with previous evaluations focusing on India (Prakash et al., 2018) and western Washington ~~State~~ state (Cao et al., 2018). In contrast, ERA5-HRES performed worse across the south-central US, where $P$ predominantly originates from small-scale, short-lived convective storms which tend to be poorly simulated by reanalyses (Adler et al., 2001; Arakawa, 2004; Ebert et al., 2007). The patterns in relative performance between IMERGHHE V05 and ERA5-HRES (Figure 4d) correspond well with those found between TMPA 3B42RT and ERA-Interim (Beck et al., 2017b, their Figure 4) and between CMORPH ~~V1.0~~ and ERA-Interim (Beck et al., 2019, their Figure 3d), suggesting that our conclusions can be generalized to other satellite and reanalysis-based $P$ datasets. Our findings suggest that topography and climate should be taken into account when choosing between satellite and reanalysis datasets. Furthermore, our results demonstrate the potential to improve continental- and global-scale $P$ datasets by merging satellite and reanalysis-based $P$ estimates ~~(Huffman et al., 1995; Xie and Arkin, 1996; Sapiano et al., 2008; Beck et al., 2017b, 2019; Zhang et al., 2018)~~(Huffman et al., 1995; Xie .

### 3.10 How well does a regional convection-permitting climate model perform?

In addition to the (quasi-)global $P$ datasets, we evaluated the performance a state-of-the-art climate simulation for the CONUS (WRF; Liu et al., 2017; Table 1). The WRF simulation has the potential to produce highly accurate $P$ estimates since it has a high 4-km resolution, which allows it to account for the influence of mesoscale orography (Doyle, 1997), and is "convection-permitting", which means it does not rely on highly uncertain convection parameterizations (Kendon et al., 2012; Prein et al., 2015). In terms of variability, WRF performed third best, being outperformed only (and very modestly) by the gauge-based CPC Unified V1.0/RT and MSWEP V2.2 datasets (Figures 1 and 2). In terms of bias, the simulation produced mixed results. WRF is the only uncorrected dataset that does not exhibit large biases over the mountainous west (Supplement Figure S2). However, large positive biases were obtained over the Great Plains region, as also found by Liu et al. (2017) using the same reference data. In terms of correlation, WRF performed worse than third-generation reanalyses (ERA-Interim, JRA-55, and NCEP-CFSR; Figure 2 and Supplement Figure S1). This is probably because WRF is forced entirely by lateral and initial boundary conditions from ERA-Interim (Liu et al., 2017), whereas the reanalyses assimilate vast amounts of *in situ* and satellite observations (Saha et al., 2010; Dee et al., 2011; Kobayashi et al., 2015). Overall, there appears to be some merit in using high-resolution, convection-permitting models to obtain climatological $P$ statistics.

## 4 Conclusions

To shed some light on the strengths and weaknesses of different precipitation ($P$) datasets and on the merit of different technological and methodological innovations, we comprehensively evaluated the performance of 26 gridded (sub-)daily $P$ datasets for the CONUS using Stage-IV gauge-radar data as reference. The evaluation was carried out at a daily ~~timescale~~ temporal and 0.1° spatial scale for the period 2008–2017 using the KGE, an objective performance metric combining correlation, bias, and variability. Our findings can be summarized as follows:

1. Across the range of KGE scores for the datasets examined the most important component is correlation (reflecting the identification of $P$ events). Of secondary importance are the $P$ totals (determining the bias score) and the distribution of $P$ intensity (affecting the variability score).

2. Among the uncorrected $P$ datasets, the (re)analyses performed better on average than the satellite-based datasets. The best performance was obtained by ECMWF's fourth-generation reanalysis ERA5-HRES, with NASA's most recent satellite-derived IMERGHHE V05 and the ensemble average ERA5-EDA coming a close equal second.

3. Among the gauge-based $P$ datasets, the best overall performance was obtained by MSWEP V2.2, followed by IMERGDF V05 and MERRA-2. The good performance of MSWEP V2.2 highlights the importance of incorporating daily gauge observations and accounting for reporting times.

4. The ~~performance~~ spread in performance among the $P$ datasets was greater in winter than in summer. The spatial variability in performance was also greater in winter for most datasets. The (re)analyses generally performed better in winter than in summer, while the opposite was the case for the satellite-based datasets.

5. The performance improvement gained after applying gauge corrections differed strongly among $P$ datasets. The largest improvements were obtained by the datasets incorporating sub-monthly gauge data (CHIRPS V2.0 and CMORPH-CRT V1.0). Several uncorrected $P$ datasets outperformed gauge-corrected ones.

6. IMERGHH V05 performed better than TMPA-3B42RT V7 for all metrics, consistent with previous studies and attributable to the many improvements implemented in the new IMERG algorithm.

7. ERA5-HRES outperformed ERA-Interim for all metrics across most of the CONUS, demonstrating the significant advances in climate and earth system modeling and data assimilation during the last decade.

8. The reduced-resolution ERA5-EDA ensemble average showed higher correlations than the high-resolution ERA5-HRES deterministic run, supporting the value of ensemble modeling. However, a side effect of the averaging is that the distribution shifted toward medium-sized events.

9. IMERGHHE V05 and ERA5-HRES showed complementary performance patterns. The former performed substantially better in regions dominated by convective storms, while the latter performed substantially better in regions of complex terrain.

10. The regional convection-permitting climate model WRF performed best among the uncorrected $P$ datasets in terms of variability. This suggest there is some merit in employing high-resolution, convection-permitting models to obtain climatological $P$ statistics.

Our findings ~~can be used as a guide~~ provide some guidance to decide which $P$ dataset should be used for a particular application. We found evidence that the relative performance of different datasets is to some degree a function of topographic complexity, climate regime, season, and rain gauge network density. Therefore, care should be taken when extrapolating our results to other regions. Additionally, results may differ when using another performance metric or when evaluating other timescales or aspects of the datasets. Similar evaluations should be carried out with other performance metrics and in other regions with ground radar networks (e.g., Australia and Europe) to verify and supplement the present findings. Of particular importance in the context of climate change is the further evaluation of $P$ extremes.

*Acknowledgements.* We gratefully acknowledge the $P$ dataset developers for producing and making available their datasets. We thank Dick Dee, an anonymous reviewer, Christa Peters-Lidard, Jelle ten Harkel, and Luca Brocca for ~~helpful comments on an earlier draft~~their thoughtful reviews of the manuscript. Hylke E. Beck was supported through IPA support from the U.S. Army Corps of Engineers' International Center for Integrated Water Resources Management (ICIWaRM), under the auspices of UNESCO. Graham P. Weedon was supported by the Joint DECC and Defra Integrated Climate Program — DECC/Defra (GA01101).

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
