# Peer review of "Daily evaluation of 26 precipitation datasets using Stage-IV gauge-radar data for the CONUS"

_Hydrology and Earth System Sciences, 2018_

## Short Comment (SC1) · 10 Oct 2018

I read this discussion paper with great interest, and we discussed some of the results in the Hydrology Working Group at the NASA Precipitation Measurement Missions Science Team meeting this week.

I applaud the authors on a comprehensive evaluation effort, and the results are useful for answering the questions posed by the authors.

From a GPM perspective, one of the critical questions not answered by this analysis is the extent to which the errors are related to detection issues or bias issues. In Tian et al., JGR, 2009, we introduced a component analysis of errors where we quantified 3 orthogonal components of error, E: Hit Error (H), Missed Precipitation (M) and False

[Figure]

Precipitation (F).

These independent components sum to the total error:

E = H - M + F

Like this study, we used Stage IV data as a reference, and in addition to producing maps of the total error and components for several products, we also found a significant seasonal cycle in these errors.

I think this reference is a critical one for this paper, and I strongly suggest that the authors dig deeper into the sources of error by computing these error components.

From Tian et al., 2009: "The relation E = H - M + F raises a critical point. It implies that it is not enough to look at the total bias E as an indicator of the performance. The three individual components H, M, and F could have larger amplitudes than the total error E, but they could cancel one another, resulting in total bias smaller than some of the components. This is especially true for M and F, which always have opposite signs. Therefore it is important to realize that the amplitude of the total bias alone is not enough to serve as a measure of the performance of a set of estimates; one needs to look at the three components as well to truly understand the error characteristics. "

Further, as can be seen from Figures 2 and 3, the errors have a pronounced seasonal cycle. An investigation of the seasonal cycle of errors would also be a useful extension of the previous work.

Reference: Tian, Y., C. D. Peters‐Lidard, J. B. Eylander, R. J. Joyce, G. J. Huffman, R. F. Adler, K. Hsu, F. J. Turk, M. Garcia, and J. Zeng (2009), Component analysis of errors in satellite‐based precipitation estimates, J. Geophys. Res., 114, D24101, doi: 10.1029/2009JD011949.

---

## Short Comment (SC2) · 25 Oct 2018

J. ten Harkel

jelle.tenharkel@wur.nl

Note to the editor and authors: As part of an introductory course to the Master programme Earth & Environment at Wageningen University, students get the assignment to review a scientific paper. Since several years, students have been reviewing papers that are in open online discussion for HESS or BGS, and they have been asked to submit their reports to the discussion in order to help the review process. While these reports are written in the form of official (invited) reviews, they were not requested for by the editor, and we leave it up to the editor and authors to use these reports to their advantage. While several students were often asked to review the same paper, this was not done with the aim to provide the authors with much extra work. We hope that these reports will positively contribute to the scientific discussion and to the quality

of papers published in HESS. This report/review was supervised by dr. Ryan Teuling (teacher within the ITEE course at Wageningen University and also associated editor with HESS).

The article of Beck et al., 2018 compares 26 different precipitation datasets and compare these datasets to one another by analysing the Kling-Gupta efficiency score (KGE score). The authors show what the limitations are of the current research performed and explains the added benefit of their research to the science community by highlighting characteristics such as the number of datasets used and the size of the geographical area (the conterminous US). Furthermore, the authors present a clear overview of the performance of these 26 datasets using a gridded KGE score for the period 2008-2017. As a reference to compare these 26 data sets to they used a radar-gauge product (Stage-IV) which has been resampled to 0.1°. They reduced systematic bias using PRISM data by matching Stage-IV long term mean to the long term mean of PRISM.

The article by Beck et al., 2018 fits the scope of the HESS Journal well. Especially the following line from the scope of HESS: "the study of the spatial and temporal characteristics of the global water resources (solid, liquid, and vapour)". It provides the reader with a helpful guide in choosing which spatiotemporal precipitation dataset they can use for specific research questions, therefore helping others in their modelling efforts. The research by Beck et al., 2018 also highlights the benefit of the newly updated precipitation datasets, showing the evolution of precipitation monitoring over the years.

The manuscript provides a good overview and evaluation of current precipitation datasets. The text is generally well-structured and concise. The conclusion of the article is in line with the evidence provided. Although the manuscript shows only limiting reasons for the performance of individual precipitation datasets, it links very well to other studies performed in this area. It can become an important reference paper for future research that uses gridded precipitation datasets. My recommendation would therefore be to publish the article after some relatively minor issues have been

addressed.

[minor issue 1] The first paragraph of the chapter 3 Results and Discussion gives the overall performance of all precipitation datasets by calculating the mean median KGE score and the KGE score components for all datasets. I wonder how useful these calculations are. In the next paragraphs the authors show how the datasets are different, so showing a mean median and making such a generalisation to start with is not useful in my opinion. I like the thought of an analysis to find the most important factor determining a high KGE score, however I wonder if for different datasets the results might be different and what the benefit is of using the KGE over normal correlation is correlation seems to be the most important factor. I would recommend leaving this paragraph out of the manuscript or clarify my concerns above. Especially clarifying the choice for KGE.

A further recommendation to analyse and assess general performance would be to include an analysis on the error associated to each dataset. Figure 2 does show box-and-whisker plots; however, no further detail is given on the underlying reasons for sometimes large whiskers. I would advise the authors to analysis this spread, instead of only focussing on the median KGE score. Analysis of this spread may prove useful in determining if specific geographic areas are underperforming compared to the median of each dataset.

[minor issue 2] As a reference to the precipitation datasets the authors used the Stage-IV dataset, which is a combination of radar and rain gauge data, they state that the dataset provides high accurate precipitation estimates. However, the authors introduce PRISM as a correction to the used Stage-IV dataset to correct for long-term mean. Again, they state that this the most accurate monthly dataset. I would like to see a better explanation of why Stage-IV is not sufficient, and the claim of the most accurate monthly dataset should be backed up with at least a reference. Plus, there should be a number showing the difference in long-term mean because at the moment it is not possible to see the difference an assess the necessity of this correction.

[Figure]

[minor issue 3] Why is the WRF dataset included according to table 2, it stopped producing data in 2013, this conflicts with the goal of the mauscript to provide a guide for the reader to choose a dataset that can be used in further research. Also, it is a mismatch to the described analysis period in paragraph 2.3, where the authors state they analyse the period 2008-2017. There are more products that mismatch this analysis period.

I would recommend that the authors explain this mismatch between available data en the chosen analysis period. Including an explanation on how this might affect the KGE scores for these specific datasets.

[minor issue 4] There are 26 data products mentioned, why is there only special focus on the dataset that have a corrected and uncorrected version in the second part of the article? Please elaborate the choice for these dataset in the introduction.

[minor issue 5] Paragraph 3.2 lines 24-31: The product SM2RAIN CCI V2 is a possible option for evaluation and correction of other datasets however the KGE of SM2RAIN CCI V2 is only 0.28, in my opinion this conflict one-another, I would like to see this further explained or removed

[minor issue 6] In the introduction, the division between the research questions 1-4 and 5-9 should become clearer, indicating that he second set of research questions is to evaluate the evolution of precipitation datasets.

Paragraph 2.3 lines 25-26 is already mentioned in on page 3 line 25.

Paragraph 3.7 line 27-18: A product MSWEP is mentioned which is completely new and doesn't add anything to the paragraph before.

In chapter 4 conclusions page 15 line 28, new things are introduced such a rain gauge density as a possible explanation, why?

In the conclusion the actual goal of the manuscript becomes clear, should be clear from the start.

[Figure]

Page 10 lines 9-11 You state that a bias is expected but this ended up not being the case, please elaborate on the expectation and on which data this expectation and conclusion are based.

Page 10 line 28, already a conclusion, can be left out here

Page 10 line 32 "suggest that its gauge-correction methodology requires re-evaluation", based on what is this statement included, please elaborate or include a reference backing up this statement.

Paragraph 3.5 mentions that IMERGHHE V05 performs better than TMPA-3B42RT V7 based on KGE scores, however figure 3a shows that in the west there are significant areas where TMPA-3B42RT V7 performs better, please indicate this in paragraph 3.5

Page 14 line 5, reference to a figure form Beck at al., 2017b), would be helpful if the figure is included in the article as a back-up to statements made in paragraph 3.8

---

## Short Comment (SC3) · 31 Oct 2018

**Short Comment**

I was pleased to read the paper by *Beck et al.* who performed a comprehensive assessment of multiple precipitation datasets over Contiguous United States (CONUS). I believe the paper is a valuable contribution. However, by reading the paper two questions raised to my mind. I believe the authors might want to address these two questions in their paper:

1) What is the value of using the Kling-Gupta Efficiency (KGE) for assessing the performance of precipitation datasets? Is it suitable for determining the products performance

for applications (e.g., flood prediction)?

2) Are the results obtained over CONUS representative of other regions? Can we generalize the obtained results?

To answer these questions, and following the final suggestions of the authors "Similar evaluations should be carried out in other regions with ground radar networks (e.g., Europe) to verify and supplement the present findings.", we tested three different satellite-based products in Europe:

a) SM2RAIN-ASCAT dataset, i.e., a recent version of an SM2RAIN-based dataset based on the application of SM2RAIN to ASCAT soil moisture product (*Brocca et al., 2017*) (apologize for self-citations). This dataset is similar to SM2RAIN-CCI V2 dataset used in *Beck et al.*

b) TMPA, the real time version of 3B42RT, i.e., TMPA-3B42RT V7 in Beck et al.

c) CMORPH, the real time version of CMORPH, i.e., CMORPH V1.0 in Beck et al.

We have considered 646 basins in Europe, and by following the same approach proposed in *Camici et al., 2018*, we tested the three satellite-based products (uncorrected) against ground-based precipitation (E-OBS dataset as reference, *Haylock et al., 2008*) for rainfall dataset assessment, and against observed discharge observations through the application of rainfall-runoff modelling.

The figure at the end of the document shows the results, in the top, for rainfall assessment by using different performance scores (KGE, R: correlation, BIAS, REL.VAR.: relative variability, RMSE: root mean square error, ubRMSE: unbiased RMSE), and in the bottom for discharge assessment by using the KGE as target score. Each dot represents a basin in which the comparison between satellite products and E-OBS is performed for rainfall assessment (basin average rainfall), and the comparison between simulated (through rainfall-runoff modelling and the three satellite rainfall datasets as
input) and observed discharge is carried out for discharge assessment. The title of each plot shows the median value of the score.

The results shown in the figure are quite interesting and illustrative of the problem in selecting a score for rainfall datasets assessment. We suppose as target the results in terms of KGE for discharge assessment shown in the bottom row.

Firstly, we underline that the results in terms of KGE for rainfall assessment (first row in the top) are not representative of the results in terms of KGE for discharge assessment. Also the use of other rainfall scores might be not suitable, with the better performance in terms of relative rankings between the products obtained by using BIAS, RMSE and ubRMSE. However, in terms of spatial assessment, each score applied to rainfall assessment seems to be not representative of discharge performances.

Secondly, the results obtained over CONUS are quite different from those we obtained here in Europe. Particularly, we want to underline the good performance of SM2RAIN-ASCAT dataset, mainly in terms of discharge assessment. This question about the representativeness of the results obtained in one region with respect to other regions.

Several other comments can be raised analysing in details the figure, but they are not suited for a short comment.

As a final comment, we want to underline that we should be cautious in saying that the results obtained over a specific region or with a specific score can be used "as a guide to choose the most suitable precipitation dataset for a particular application.". We believe that more research is still needed and a significant effort linking satellite, meteorological and hydrological community is needed for a robust assessment of the precipitation datasets for hydrological applications.

**References**

Brocca, L., Crow, W.T., Ciabatta, L., Massari, C., de Rosnay, P., Enenkel, M.,
Hahn, S., Amarnath, G., Camici, S., Tarpanelli, A., Wagner, W. (2017). A review of the applications of ASCAT soil moisture products. IEEE Journal of Selected Topics in Applied Earth Observations and Remote Sensing, 10(5), 2285-2306, doi:10.1109/JSTARS.2017.2651140.

Camici, S., Ciabatta, L., Massari, C., Brocca, L. (2018). How reliable are satellite precipitation estimates for driving hydrological models: a verification study over the Mediterranean area. Journal of Hydrology, 563, 950-961, doi:10.1016/j.jhydrol.2018.06.067.

Haylock, M. R., Hofstra, N., Klein Tank, A. M. G., Klok, E. J., Jones, P. D., New, M. (2008). A European daily high-resolution gridded data set of surface temperature and precipitation for 1950–2006. Journal of Geophysical Research: Atmospheres, 113(D20), doi:10.1029/2008JD010201.

---

## Referee Comment (RC1) · D. Dee (Referee) · 11 Nov 2018

This paper presents an evaluation of 26 (near-)global precipitation datasets, using as a reference the NCEP Stage-IV dataset derived from radar and rain gauge data. All datasets are ranked in terms of statistical fit (correlation, bias and variability) of daily accumulations at 0.1 degree resolution over the conterminous US for the period 2008-2017. Datasets are divided in two categories: those that have been explicitly corrected to gauge data and those that have not. They are further separated based on the main sources of data used. Two very useful tables list the main characteristics and primary reference for all datasets used in this study.

Results of the evaluation are usefully summarised in two figures. Discussion of results is framed in terms of 9 topics, and conclusions are presented as a list of short statements. The text is kept relatively short, relying on an extensive list of references covering related studies and reviews.

I really like the approach taken by the authors in summarising the datasets and evaluation results to ensure that this paper remains readable and focussed, yet does justice to the complexity of precipitation datasets and the evaluation of their quality and usefulness. One could argue that the list of 26 datasets is far from complete, however the selection covers the most-used datasets and also represents well the different methodologies and data sources available. The statistical evaluation is simple yet addresses the key measures that one would look at first in any study such as this. (Having said that, it would be very interesting to see correlations on the hourly timescale for those datasets with sufficient temporal resolution.) The topics for discussion are phrased as questions that follow naturally from the statistical evaluation. I think this also works very well.

Near the end of the paper the authors point out that their findings can be used to help users decide which dataset should be used for their particular application. I think this is a very important point, especially since data on precipitation (and several other climate parameters) are increasingly used by non-specialists to support planning and decision making, potentially with significant implications for society. It is very difficult to make a study such as this accessible to those users - in my opinion the authors have done this very well.

HESSD

---

## Referee Comment (RC2) · Anonymous Referee #2 · 5 Dec 2018

This study compares 26 precipitation datasets with respect to the Stage-IV product at 0.1 degree resolution over the CONUS and at the daily time scale for the period 2008–2017. The Kling-Gupta efficiency is primarily used to rank the datasets, emphasizing the correlation component. The importance of gauge reporting times in daily gauge corrections is highlighted. Examples comparisons between product versions, satellite versus reanalyzes, deterministic versus ensemble reanalyzes are provided.

The topic fits the scope of the journal as it presents an overview of a selection of available precipitation products. The paper is easy to follow and the methodology is clear. However the limitations of such exercise need to be better highlighted. The paper would be suitable for publication after the following comments are addressed:

1. While the use of a score like KGE is convenient for intercomparison exercises, it

must be recalled that such an integrative metric only partially depicts actual performances of precipitation products. This is all the more true since the median KGE values are primarily used in this analysis, which further prevents a detailed assessment. Errors are multi-faceted and scale dependent. As the authors point out KGE in this context emphasizes correlation, which tends to bias the assessment by favoring products designed to correctly capture the timing of daily events such as MSWEP. Other precipitation aspects such as daily totals are of primary importance for hydrological applications. The use of another metric could generate different conclusions. As stated by Gupta et al. (2009) who initially proposed the KGE score, "the primary purpose of this study is not to present an improved measure of model performance", but "suggest possible ways forward that may move us towards an improved and diagnostically meaningful approach to model performance evaluation and identification". It is recommended to explicit the relevance of the KGE with respect to the assessment purpose.

2. Stage IV is not a homogeneous precipitation product over the CONUS. Its generation varies across River Forecast Centers, e.g. it relies more heavily on PRISM in the Western U.S. A homogeneous reference is ideal for such assessment of precipitation products, and the lack thereof should be mentioned.

3. No seasonal dependency of the performances is reported in this analysis, although it is an important factor. See e.g. Gebregiorgis et al (2018) for a comparison between TMPA and IMERG over the CONUS. For example performances during the winter season are of significant interest especially for snow conditions. Accurate solid precipitation estimation is of primary importance for applications such as water resources management. Can the authors comment this aspect? Reference: Gebregiorgis et al., 2018: To what extent is the day 1 GPM IMERG satellite precipitation estimate improved as compared to TRMM TMPA‐RT?. Journal of Geophysical Research: Atmospheres, 123, 1694–1707. https://doi.org/10.1002/2017JD027606

4. Other important aspects of precipitation such as occurrence or extremes are not

assessed in this study, although they are of primary importance for an array of applications of precipitation products. Can the authors comment this aspect?

5. Precipitation (solid, liquid and mixed phase) has a large spatial and temporal variability. This scale dependency limits the representativeness of this intercomparison exercise to the daily time scale. Applications such as flash flood forecasting would require an evaluation at finer time scale. Can the authors comment this aspect?

6. For the above reasons it cannot be stated as in the abstract and in the conclusion that "Our findings can be used as a guide to choose the most suitable dataset for a particular application". Applications require a refined and more detailed assessment than the one proposed in this study. Please modify this statement.

---

## Author Comment (AC1) · 19 Dec 2018

Dear Dr. ten Veldhuis,

We hereby provide responses to the reviewer comments for our manuscript entitled "*Daily evaluation of 26 precipitation datasets using Stage-IV gauge-radar data for the CONUS*" in green font below. We provide responses to both referee comments as well as to the three short comments. The revised manuscript is attached.

We have made several changes which have resulted in a much better paper. Most importantly, we have added a comparison between the performance in summer and winter (see Section 3.4), in response to comments by multiple reviewers. In addition, we have refined many statements and added several references.

We would like to sincerely thank you for handling the manuscript in such a prompt and efficient manner.

Sincerely,

Hylke Beck (on behalf of all co-authors)

**Reviewer #1**

D. Dee

In the interest of transparency, it is important to note that the reviewer and one of the co-authors work at the same institution (the European Centre for Medium-Range Weather Forecasts — ECMWF) albeit in different departments. Part of the reviewer's responsibility is to lead the development of re-analysis, while the co-author is responsible for the operational production of the re-analysis and all other forecasts produced by ECMWF.

This paper presents an evaluation of 26 (near-)global precipitation datasets, using as a reference the NCEP Stage-IV dataset derived from radar and rain gauge data. All datasets are ranked in terms of statistical fit (correlation, bias and variability) of daily accumulations at 0.1 degree resolution over the conterminous US for the period 2008-2017. Datasets are divided in two categories: those that have been explicitly corrected to gauge data and those that have not. They are further separated based on the main sources of data used. Two very useful tables list the main characteristics and primary reference for all datasets used in this study.

Results of the evaluation are usefully summarised in two figures. Discussion of results is framed in terms of 9 topics, and conclusions are presented as a list of short statements. The text is kept relatively short, relying on an extensive list of references covering related studies and reviews.

I really like the approach taken by the authors in summarising the datasets and evaluation results to ensure that this paper remains readable and focussed, yet does justice to the complexity of precipitation datasets and the evaluation of their quality and usefulness. One could argue that the list of 26 datasets is far from complete, however the selection covers the most-used datasets and also represents well the different methodologies and data sources available. The statistical evaluation is simple yet addresses the key measures that one would look at first in any study such as this. (Having said that, it would be very interesting to see correlations on the hourly timescale for those datasets with sufficient temporal resolution.) The topics for discussion are phrased as questions that follow naturally from the statistical evaluation. I think this also works very well.

We would like to thank Dr. Dee for his review and the compliments. Our intention was to produce a thorough yet easy-to-follow paper and we are glad to have accomplished this.

Regarding the hourly scale, we agree that such an evaluation would be an interesting follow-up study. While we have performed an evaluation at the 3-hourly scale in Beck et al. (2019), we concur that a more comprehensive evaluation is necessary.

Near the end of the paper the authors point out that their findings can be used to help users decide which dataset should be used for their particular application. I think this is a very important point, especially since data on precipitation (and several other climate parameters) are increasingly used by non-specialists to support planning and decision making, potentially with significant implications

for society. It is very difficult to make a study such as this accessible to those users - in my opinion the authors have done this very well.

We completely agree. We think that the large number of precipitation datasets currently available can be overwhelming to people new to the field, and hope that the current study provides some answers to frequently asked questions.

[revised manuscript text omitted]

---

## Author Comment (AC2) · 19 Dec 2018

**Reviewer #2**

This study compares 26 precipitation datasets with respect to the Stage-IV product at 0.1 degree resolution over the CONUS and at the daily time scale for the period 2008–2017. The Kling-Gupta efficiency is primarily used to rank the datasets, emphasizing the correlation component. The importance of gauge reporting times in daily gauge corrections is highlighted. Examples comparisons between product versions, satellite versus reanalyzes, deterministic versus ensemble reanalyzes are provided.

The topic fits the scope of the journal as it presents an overview of a selection of available precipitation products. The paper is easy to follow and the methodology is clear. However the limitations of such exercise need to be better highlighted. The paper would be suitable for publication after the following comments are addressed:

We want to thank the reviewer for their thorough assessment of the paper. We appreciate the comment that the paper is "*easy to follow*" as this is exactly what we were aiming for.

1. While the use of a score like KGE is convenient for intercomparison exercises, it must be recalled that such an integrative metric only partially depicts actual performances of precipitation products. This is all the more true since the median KGE values are primarily used in this analysis, which further prevents a detailed assessment. Errors are multi-faceted and scale dependent. As the authors point out KGE in this context emphasizes correlation, which tends to bias the assessment by favoring products designed to correctly capture the timing of daily events such as MSWEP. Other precipitation aspects such as daily totals are of primary importance for hydrological applications. The use of another metric could generate different conclusions. As stated by Gupta et al. (2009) who initially proposed the KGE score, "the primary purpose of this study is not to present an improved measure of model performance", but "suggest possible ways forward that may move us towards an improved and diagnostically meaningful approach to model performance evaluation and identification". It is recommended to explicit the relevance of the KGE with respect to the assessment
Purpose.

The KGE is indeed an integrative metric that does not immediately reveal which aspects of the datasets are different. We therefore also show detailed results for the components of the KGE (correlation, bias, variability ratio) for all 26 datasets. See Figure 2 of the main paper and Supplementary Figures S1, S2, and S3.

We agree that "*daily totals are of primary importance for hydrological applications*" and this is exactly what the bias component of the KGE evaluates. Summary results for the bias are presented in Figure 2 of the main paper with detailed maps for all datasets provided in Supplementary Figure S2.

We have chosen the KGE for the present study since it is an objective metric that evaluates the most important aspects of data series, as explained in Section 2.3. For this reason, the metric is becoming more and more popular in different fields of science.

Nevertheless, we are in complete agreement with the reviewer that "*the use of another metric could generate different conclusions*" and we already explicitly state this in the paper in the last paragraph of the Conclusions: "*results may differ when using another performance metric or when evaluating other timescales or aspects of the datasets.*" To emphasize the importance of follow-up research using different performance metrics, we have modified the last sentence in the Conclusions from: "*Similar evaluations should be carried out in other regions with ground radar networks (e.g., Europe) to verify and supplement the present findings*" to "*Similar evaluations should be carried out with other performance metrics and in other regions with ground radar networks (e.g., Australia and Europe) to verify and supplement the present findings.*"

2. Stage IV is not a homogeneous precipitation product over the CONUS. Its generation varies across River Forecast Centers, e.g. it relies more heavily on PRISM in the Western U.S. A homogeneous reference is ideal for such assessment of precipitation products, and the lack thereof should be mentioned.

Thanks for the comment. We have added the following sentence including three references: "*The Stage-IV dataset is a mosaic of regional analyses produced by 12 CONUS River Forecast Centers (RFCs) and is thus subject to the gauge correction and quality control performed at each individual RFC (Westrick et al., 1999; Smalley et al., 2014; Eldardiry et al., 2017).*" The poorer performance of Stage-IV (as well as the evaluated precipitation datasets) over the western CONUS is discussed in Section 3.2 and is attributed to "*the more complex topography and greater spatiotemporal heterogeneity of P in the west*".

3. No seasonal dependency of the performances is reported in this analysis, although it is an important factor. See e.g. Gebregiorgis et al (2018) for a comparison between TMPA and IMERG over the CONUS. For example performances during the winter season are of significant interest especially for snow conditions. Accurate solid precipitation estimation is of primary importance for applications such as water resources management. Can the authors comment this aspect? Reference: Gebregiorgis et al., 2018: To what extent is the day 1 GPM IMERG satellite precipitation estimate improved as compared to TRMM TMPAâ˘Aˇ RRT?. Journal of Geophysical Research: Atmospheres, 123, 1694–1707. https://doi.org/10.1002/2017JD027606

We have added a new question to the paper which compares the performance of the datasets in summer versus winter (see Section 3.4), providing some very interesting insights. In addition, we cite Gebregiorgis et al. (2018) in the revised paper. Thank you for pointing us to this very useful study.

4. Other important aspects of precipitation such as occurrence or extremes are not assessed in this study, although they are of primary importance for an array of applications of precipitation products. Can the authors comment this aspect?

We agree that precipitation extremes are important for a host of hydrological applications. However, we do not agree that precipitation extremes have not been assessed in the present study, as the

correlation (the first component of the KGE) is primarily sensitive to the dynamics of the largest values (i.e., the extremes). In addition, the variability ratio (the third component of the KGE) reflects the distribution of the daily values and is thus sensitive to the magnitude of extremes.

5. Precipitation (solid, liquid and mixed phase) has a large spatial and temporal variability. This scale dependency limits the representativeness of this intercomparison exercise to the daily time scale. Applications such as flash flood forecasting would require an evaluation at finer time scale. Can the authors comment this aspect?

Many of the precipitation datasets evaluated in this study have a daily temporal resolution and we therefore focused on the daily time scale. We agree that a daily time scale is not suitable for short-range flash flood forecasting. The value of an evaluation at a finer temporal resolution has also been brought up by Reviewer #1. We have already performed an evaluation at the 3-hourly scale using Stage-IV (see Beck et al., 2019), but agree that a more thorough evaluation is needed and emphasize the need for more research in the Conclusions.

6. For the above reasons it cannot be stated as in the abstract and in the conclusion that "Our findings can be used as a guide to choose the most suitable dataset for a particular application". Applications require a refined and more detailed assessment than the one proposed in this study. Please modify this statement.

This comment is in contradiction with the final comment by Reviewer #1. Nevertheless, we appreciate the suggestion and have modified the statements in both the abstract and the conclusion. The old statement in the Conclusions read: *"Our findings can be used as a guide to choose the most suitable P dataset for a particular application."* The revised statement in the Conclusions reads: *"Our findings provide some guidance to decide which P dataset should be used for a particular application."* The statement in the abstract has been revised in a similar fashion.

---

## Author Comment (AC3) · 19 Dec 2018

C. Peters-Lidard

I read this discussion paper with great interest, and we discussed some of the results in the Hydrology Working Group at the NASA Precipitation Measurement Missions Science Team meeting this week.

I applaud the authors on a comprehensive evaluation effort, and the results are useful for answering the questions posed by the authors.

We thank Dr. Peters-Lidard for her comment on our paper.

From a GPM perspective, one of the critical questions not answered by this analysis is the extent to which the errors are related to detection issues or bias issues. In Tian et al., JGR, 2009, we introduced a component analysis of errors where we quantified 3 orthogonal components of error, E: Hit Error (H), Missed Precipitation (M) and False Precipitation (F).

We provide individual scores for correlation, bias, and variability ratio for all 26 datasets (see Figures 1 and 2 in the main paper and S1, S2, and S3 in the Supplement). We do not agree that *"one of the critical questions not answered by this analysis is the extent to which the errors are related to detection issues or bias issues"* as detection issues are primarily reflected in the correlation values and bias issues in the bias values. We therefore certainly do make a distinction between detection and bias issues in the paper.

The drawback of a "hit, miss, false alarm" type of evaluation is that (1) it involves the (somewhat arbitrary) selection of a precipitation threshold for event identification and (2) the agreement in terms of magnitude beyond this threshold is not further evaluated. Conversely, the correlation does account for differences in magnitude beyond this threshold.

These independent components sum to the total error:
$E = H - M + F$

Like this study, we used Stage IV data as a reference, and in addition to producing maps of the total error and components for several products, we also found a significant seasonal cycle in these errors.

We have added a new question to the paper which compares the performance of the datasets between summer and winter (see Section 3.4 of the revised paper).

I think this reference is a critical one for this paper, and I strongly suggest that the authors dig deeper into the sources of error by computing these error components.

We have added a reference this very interesting paper. For reasons stated above and because it would make the paper considerably less concise we are not in favor of adding a "hit, miss, false alarm" analysis to the study.

From Tian et al., 2009: "The relation E = H - M + F raises a critical point. It implies that it is not enough to look at the total bias E as an indicator of the performance. The three individual components H, M, and F could have larger amplitudes than the total error E, but they could cancel one another, resulting in total bias smaller than some of the components. This is especially true for M and F, which always have opposite signs. Therefore it is important to realize that the amplitude of the total bias alone is not enough to serve as a measure of the performance of a set of estimates; one needs to look at the three components as well to truly understand the error characteristics. "

Thank you for the quote. However, we do not only focus on the bias in our study. We also focus on correlation and variability ratio. The correlation reflects the performance of the datasets in terms of event detection, as mentioned earlier in this response.

Further, as can be seen from Figures 2 and 3, the errors have a pronounced seasonal cycle. An investigation of the seasonal cycle of errors would also be a useful extension of the previous work.

Agreed. We have added a new question to the paper in which we compare the performance of the datasets between summer and winter (see Section 3.4 of the revised paper).

Reference: Tian, Y., C. D. Petersˇ˘AˇRLidard, J. B. Eylander, R. J. Joyce, G. J. Huffman, R. F. Adler, K. Hsu, F. J. Turk, M. Garcia, and J. Zeng (2009), Component analysis of errors in satelliteˆ˘AˇRbased precipitation estimates, J. Geophys. Res., 114, D24101, doi: 10.1029/2009JD011949.

---

## Author Comment (AC4) · 19 Dec 2018

J. ten Harkel

Note to the editor and authors: As part of an introductory course to the Master programme Earth & Environment at Wageningen University, students get the assignment to review a scientific paper. Since several years, students have been reviewing papers that are in open online discussion for HESS or BGS, and they have been asked to submit their reports to the discussion in order to help the review process. While these reports are written in the form of official (invited) reviews, they were not requested for by the editor, and we leave it up to the editor and authors to use these reports to their advantage. While several students were often asked to review the same paper, this was not done with the aim to provide the authors with much extra work. We hope that these reports will positively contribute to the scientific discussion and to the quality of papers published in HESS. This report/review was supervised by dr. Ryan Teuling (teacher within the ITEE course at Wageningen University and also associated editor with HESS).

We would like to congratulate Wageningen University for this initiative — peer review is a fundamental aspect of science and scientific publishing and adding it to the curriculum is an excellent idea.

The article of Beck et al., 2018 compares 26 different precipitation datasets and compare these datasets to one another by analysing the Kling-Gupta efficiency score (KGE score). The authors show what the limitations are of the current research performed and explains the added benefit of their research to the science community by highlighting characteristics such as the number of datasets used and the size of the geographical area (the conterminous US). Furthermore, the authors present a clear overview of the performance of these 26 datasets using a gridded KGE score for the period 2008-2017. As a reference to compare these 26 data sets to they used a radar-gauge product (Stage-IV) which has been resampled to $0.1°$. They reduced systematic bias using PRISM data by matching Stage-IV long term mean to the long term mean of PRISM.

The article by Beck et al., 2018 fits the scope of the HESS Journal well. Especially the following line from the scope of HESS: "the study of the spatial and temporal characteristics of the global water resources (solid, liquid, and vapour)". It provides the reader with a helpful guide in choosing which spatiotemporal precipitation dataset they can use for specific research questions, therefore helping others in their modelling efforts. The research by Beck et al., 2018 also highlights the benefit of the newly updated precipitation datasets, showing the evolution of precipitation monitoring over the years. The manuscript provides a good overview and evaluation of current precipitation datasets. The text is generally well-structured and concise. The conclusion of the article is in line with the evidence provided. Although the manuscript shows only limiting reasons for the performance of individual precipitation datasets, it links very well to other studies performed in this area. It can become an important reference paper for future research that uses gridded precipitation datasets. My recommendation would therefore be to publish the article after some relatively minor issues have been addressed.

Thank you for the thorough review which has helped us to improve the paper.

[minor issue 1] The first paragraph of the chapter 3 Results and Discussion gives the overall performance of all precipitation datasets by calculating the mean median KGE score and the KGE score components for all datasets. I wonder how useful these calculations are. In the next paragraphs the authors show how the datasets are different, so showing a mean median and making such a generalisation to start with is not useful in my opinion. I like the thought of an analysis to find the most important factor determining a high KGE score, however I wonder if for different datasets the results might be different and what the benefit is of using the KGE over normal correlation is correlation seems to be the most important factor. I would recommend leaving this paragraph out of the manuscript or clarify my concerns above. Especially clarifying the choice for KGE.

We calculate mean scores for the KGE and its components to demonstrate that, among the three components, the correlation is on average the "worst" and therefore exerts the dominant influence on the final KGE scores. The calculation of the mean scores is necessary to make this point. We hope this clarifies the issue.

A further recommendation to analyse and assess general performance would be to include an analysis on the error associated to each dataset. Figure 2 does show box-and-whisker plots; however, no further detail is given on the underlying reasons for sometimes large whiskers. I would advise the authors to analysis this spread, instead of only focussing on the median KGE score. Analysis of this spread may prove useful in determining if specific geographic areas are underperforming compared to the median of each dataset.

The data underlying the box-and-whisker plots shown in Figure 2 are presented in Figure 1. It is difficult to ascertain the reasons why a certain dataset performs as it does, but we have made every effort to do so.

[minor issue 2] As a reference to the precipitation datasets the authors used the Stage-IV dataset, which is a combination of radar and rain gauge data, they state that the dataset provides high accurate precipitation estimates. However, the authors introduce PRISM as a correction to the used Stage-IV dataset to correct for long-term mean. Again, they state that this the most accurate monthly dataset. I would like to see a better explanation of why Stage-IV is not sufficient, and the claim of the most accurate monthly dataset should be backed up with at least a reference. Plus, there should be a number showing the difference in long-term mean because at the moment it is not possible to see the difference an assess the necessity of this correction.

We have added a sentence stating that the PRISM dataset has been used as reference in several precipitation dataset evaluation studies and provide three examples: *"It is generally considered the most accurate monthly P dataset available for the US and has been used as reference in numerous studies (e.g., Mizukami et al., 2012; Prat and Nelson, 2015; Liu et al., 2017)."*

[minor issue 3] Why is the WRF dataset included according to table 2, it stopped producing data in 2013, this conflicts with the goal of the mauscript to provide a guide for the reader to choose a dataset that can be used in further research. Also, it is a mismatch to the described analysis period in paragraph 2.3, where the authors state they analyse the period 2008-2017. There are more products that mismatch this analysis period.

The fact that a precipitation dataset is only available for the past does not mean that the dataset cannot be used for research. The WRF dataset, for example, can be used to study the impact of climate change on precipitation patterns in the US.

I would recommend that the authors explain this mismatch between available data en the chosen analysis period. Including an explanation on how this might affect the KGE scores for these specific datasets.

This is indeed a (minor) drawback of the study but one that is impossible to overcome due to the different start and end times of the datasets. This is however unlikely to influence the results since even the shortest period of record of four years (for IMERGHHE V05) is still more than sufficient to calculate robust performance statistics.

[minor issue 4] There are 26 data products mentioned, why is there only special focus on the dataset that have a corrected and uncorrected version in the second part of the article? Please elaborate the choice for these dataset in the introduction.

We do not fully understand the question. However, the distinction between uncorrected and corrected datasets was necessary to avoid unfair comparisons.

[minor issue 5] Paragraph 3.2 lines 24-31: The product SM2RAIN CCI V2 is a possible option for evaluation and correction of other datasets however the KGE of SM2RAIN CCI V2 is only 0.28, in my opinion this conflict one-another, I would like to see this further explained or removed.

We appreciate the suggestion but the two studies we cite clearly demonstrate that SM2RAIN makes it possible to evaluate and correct other precipitation datasets. We refer to those studies for more information.

[minor issue 6] In the introduction, the division between the research questions 1-4 and 5-9 should become clearer, indicating that he second set of research questions is to evaluate the evolution of precipitation datasets.

Questions 7 (now 8) and 9 (now 10) are not related to the evolution of precipitation datasets. The proposed distinction is thus not entirely valid.

Paragraph 2.3 lines 25-26 is already mentioned in on page 3 line 25.

Good suggestion. We have corrected this.

Paragraph 3.7 line 27-18: A product MSWEP is mentioned which is completely new and doesn't add anything to the paragraph before.

We agree and have removed the sentence in question.

In chapter 4 conclusions page 15 line 28, new things are introduced such a rain gauge density as a possible explanation, why?

This statement serves to bring two additional factors to light that should be considered when attempting to generalize the results of gauge-corrected precipitation datasets to other regions.

In the conclusion the actual goal of the manuscript becomes clear, should be clear from the start.

The last paragraph of the Introduction clearly lists the objective of the paper: *"We shed light on the strengths and weaknesses of different P datasets and on the merit of different technological and methodological innovations by addressing ten pertinent questions."*

Page 10 lines 9-11 You state that a bias is expected but this ended up not being the case, please elaborate on the expectation and on which data this expectation and conclusion are based.

We expected a bias to be present *"since PRISM, the dataset used to correct systematic biases in Stage-IV (see Section 2.2), lacks explicit gauge undercatch corrections (Daly et al., 2008)."*

Page 10 line 28, already a conclusion, can be left out here

The statement that *"the results found here for the CONUS do not necessarily directly generalize to other regions"* is important and does not feel out of place to us.

Page 10 line 32 "suggest that its gauge-correction methodology requires re-evaluation", based on what is this statement included, please elaborate or include a reference backing up this statement.

The fact that *"GSMaP-Std Gauge V7 shows a large positive bias in the west (Supplement Figure S2)"* suggests that its algorithm exhibits serious issues. We are not aware of other studies reporting about this issue and hence cannot provide a reference.

Paragraph 3.5 mentions that IMERGHHE V05 performs better than TMPA-3B42RT V7 based on KGE scores, however figure 3a shows that in the west there are significant areas where TMPA-3B42RT V7 performs better, please indicate this in paragraph 3.5

Thanks for the comment. We have added the following: *"In the west, however, there are still some small regions over which TMPA-3B42RT V7 performs better (Figure 4a)."*

Page 14 line 5, reference to a figure form Beck at al., 2017b), would be helpful if the figure is included in the article as a back-up to statements made in paragraph 3.8

We are not in favor of repeating the results of previous studies as this would make the paper significantly less concise.

---

## Author Comment (AC5) · 19 Dec 2018

**Short comment**

L. Brocca

We would like to thank Dr. Brocca for the valuable comment and in the interest of transparency we highlight that we have interacted with Dr. Brocca beforehand, which led to the removal of the SM2RAIN-SMAP data set from the evaluation.

I was pleased to read the paper by Beck et al. who performed a comprehensive assessment of multiple precipitation datasets over Contiguous United States (CONUS). I believe the paper is a valuable contribution. However, by reading the paper two questions raised to my mind. I believe the authors might want to address these two questions in their paper:

1) What is the value of using the Kling-Gupta Efficiency (KGE) for assessing the performance of precipitation datasets? Is it suitable for determining the products performance for applications (e.g., flood prediction)?

As mentioned in the paper, the KGE is an objective and widely used performance metric that combines three fully independent performance metrics: correlation, bias, and variability ratio. The metric thus evaluates the most important aspects of the precipitation time series and is therefore *"suitable for determining the products performance for applications (e.g., flood prediction)"*. However, it is important to present the results for all three components in addition to the KGE scores, as we have done in our paper.

2) Are the results obtained over CONUS representative of other regions? Can we generalize the obtained results?

The comprehensive global-scale evaluation by Beck et al. (2017) showed similar results between Europe and the CONUS, suggesting that the results of our new study are generalizable to Europe. However, some of the datasets used in the comparison (i.e., ERA5) use additional data in their assimilation over the CONUS and hence transferability of the results may be challenging. Therefore, we emphasize the need for more research to verify and supplement the present findings in the last paragraph of the Conclusions.

To answer these questions, and following the final suggestions of the authors "Similar evaluations should be carried out in other regions with ground radar networks (e.g., Europe) to verify and supplement the present findings.", we tested three different satellite-based products in Europe:

a) SM2RAIN-ASCAT dataset, i.e., a recent version of an SM2RAIN-based dataset based on the application of SM2RAIN to ASCAT soil moisture product (Brocca et al.,2017) (apologize for self-citations). This dataset is similar to SM2RAIN-CCI V2 dataset used in Beck et al.

b) TMPA, the real time version of 3B42RT, i.e., TMPA-3B42RT V7 in Beck et al.

c) CMORPH, the real time version of CMORPH, i.e., CMORPH V1.0 in Beck et al.

We have considered 646 basins in Europe, and by following the same approach proposed in Camici et al., 2018, we tested the three satellite-based products (uncorrected) against ground-based precipitation (E-OBS dataset as reference, Haylock et al., 2008) for rainfall dataset assessment, and against observed discharge observations through the application of rainfall-runoff modelling.

The figure at the end of the document shows the results, in the top, for rainfall assessment by using different performance scores (KGE, R: correlation, BIAS, REL.VAR.: relative variability, RMSE: root mean square error, ubRMSE: unbiased RMSE), and in the bottom for discharge assessment by using the KGE as target score. Each dot represents a basin in which the comparison between satellite products and E-OBS is performed for rainfall assessment (basin average rainfall), and the comparison between simulated (through rainfall-runoff modelling and the three satellite rainfall datasets as input) and observed discharge is carried out for discharge assessment. The title of each plot shows the median value of the score.

The results shown in the figure are quite interesting and illustrative of the problem in selecting a score for rainfall datasets assessment. We suppose as target the results in terms of KGE for discharge assessment shown in the bottom row.

Firstly, we underline that the results in terms of KGE for rainfall assessment (first row in the top) are not representative of the results in terms of KGE for discharge assessment. Also the use of other rainfall scores might be not suitable, with the better performance in terms of relative rankings between the products obtained by using BIAS, RMSE and ubRMSE. However, in terms of spatial assessment, each score applied to rainfall assessment seems to be not representative of discharge performances.

First, in his discharge assessment, Dr. Brocca relies on a single performance metric (KGE). We want to caution against relying on a single performance metric as this leaves room for speculation when interpreting the results.

Second, the transformation from rainfall to runoff is a complex and highly non-linear process. All rainfall-runoff models (both physical and conceptual) are simplified representations of reality and therefore require calibration. Evaporation can easily compensate for many kinds of errors in precipitation, particularly since it is such a poorly observed variable. Channel routing may also mask certain errors in precipitation, due to its smoothing effect. It is possible that the employed rainfall-runoff model provides better discharge simulations when rainfall peaks are underestimated, as is the case for SM2RAIN-ASCAT. The better discharge simulation performance does, however, not necessarily mean that SM2RAIN-ASCAT is a more accurate precipitation dataset. In our opinion, the evaluation using EOBS is more informative in this regard.

Third, the analysis presented in the short comment includes only three precipitation datasets. We feel this is insufficient to draw robust conclusions about the differences in ranking between the rainfall and discharge assessments, particularly since SM2RAIN and CMORPH exhibit rather

peculiar issues (strong underestimation of peaks and winter precipitation, respectively; Beck et al., 2017).

Fourth, regarding the RMSE metric, we argue that it should be avoided for the evaluation of precipitation datasets at (sub-)daily time scales as it can yield misleading results. This is due to the high skewness of the precipitation distribution and the prevalence of temporal mismatches between estimated and observed precipitation peaks. We will illustrate this with the following example. Take two precipitation products: (1) the original ERA-Interim (i.e., drizzly and underestimated peaks) and (2) an improved version of ERA-Interim with a perfect cumulative distribution function (CDF; i.e., same temporal dynamics as the original but less drizzly and no peak underestimation). For many locations, the improved ERA-Interim would yield a worse (i.e., higher) RMSE score because the higher peaks result in larger RMSE values when there are temporal mismatches between estimated and observed peaks, which is frequently the case. The KGE does not suffer from this problem, and would give a better (i.e., higher) score for the improved ERA-Interim. The RMSE values presented by Dr. Brocca in his short comment are likely affected by this issue and should therefore be interpreted with caution.

Secondly, the results obtained over CONUS are quite different from those we obtained here in Europe. Particularly, we want to underline the good performance of SM2RAIN-ASCAT dataset, mainly in terms of discharge assessment. This question about the representativeness of the results obtained in one region with respect to other regions. Several other comments can be raised analysing in details the figure, but they are not suited for a short comment.

As mentioned before, the global-scale evaluation by Beck et al. (2017) shows very similar results between Europe and the CONUS for SM2RAIN-CCI (which is very similar to SM2RAIN-ASCAT) and the other precipitation datasets, suggesting that the results of the current study are generalizable to Europe. Nevertheless, we recognize the need for more research, as explicitly mentioned in the last paragraph of the Conclusions.

As a final comment, we want to underline that we should be cautious in saying that the results obtained over a specific region or with a specific score can be used "as a guide to choose the most suitable precipitation dataset for a particular application." We believe that more research is still needed and a significant effort linking satellite, meteorological and hydrological community is needed for a robust assessment of the precipitation datasets for hydrological applications.

Thanks for the comment. We agree and have softened the statement as follows: *"Our findings provide some guidance to decide which P dataset should be used for a particular application."*

References

Brocca, L., Crow, W.T., Ciabatta, L., Massari, C., de Rosnay, P., Enenkel, M. Hahn, S., Amarnath, G., Camici, S., Tarpanelli, A., Wagner, W. (2017). A review of the applications of ASCAT soil moisture products. IEEE Journal of Selected Topics in Applied Earth Observations and Remote Sensing, 10(5), 2285-2306, doi:10.1109/JSTARS.2017.2651140.

Camici, S., Ciabatta, L., Massari, C., Brocca, L. (2018). How reliable are satellite precipitation estimates for driving hydrological models: a verification study over the Mediterranean area. Journal of Hydrology, 563, 950-961, doi:10.1016/j.jhydrol.2018.06.067.

Haylock, M. R., Hofstra, N., Klein Tank, A. M. G., Klok, E. J., Jones, P. D., New, M. (2008). A European daily high-resolution gridded data set of surface temperature and precipitation for 1950–2006. Journal of Geophysical Research: Atmospheres, 113(D20), doi:10.1029/2008JD010201.

---

## Author Comment (AC6) · 26 Dec 2018

The comment was uploaded in the form of a supplement:
https://www.hydrol-earth-syst-sci-discuss.net/hess-2018-481/hess-2018-481-AC6-supplement.pdf